# METASPATIAL: REINFORCING 3D SPATIAL REASONING IN VLMS FOR THE METAVERSE

**Zhenyu Pan**
Department of Computer Science
Northwestern University
`zhenyupan@u.northwestern.edu`

**Han Liu**
Department of Computer Science
Northwestern University
`hanliu@northwestern.edu`

## ABSTRACT

We present **MetaSpatial**, the first reinforcement learning (RL) framework for enhancing 3D spatial reasoning in vision-language models (VLMs), enabling real-time 3D scene layout generation without post-processing. MetaSpatial addresses two key challenges: (i) the need for extensive post-processing, as existing VLMs lack inherent 3D spatial reasoning to generate realistic layouts; and (ii) the inefficiency of supervised fine-tuning (SFT) for layout generation due to scarcity of perfect annotations. Our core contribution is the 3D Spatial Policy Optimization (**3D-SPO**) algorithm, which incorporates physics-aware modulation into advantage estimates at the object level and trajectory-level reward from a training-only multi-turn refinement pipeline. This design enhances temporal credit assignment and encourages spatially consistent policy learning. Empirical evaluations across models of varying scales demonstrate that MetaSpatial improves spatial coherence, physical plausibility, and formatting stability, leading to more realistic and functionally coherent object placements applicable to metaverse environments.

## 1 INTRODUCTION

This work introduces **MetaSpatial**, the first RL-based framework designed to enhance the 3D spatial reasoning capabilities of VLMs for 3D scene layout generation. It addresses two key challenges in existing methods: (1) the need for extensive post-processing, due to the lack of internalized 3D spatial reasoning in VLMs, which limits their ability to generate realistic and coherent scene layouts; and (2) the inherent limitations of SFT, which assumes a single "correct" layout. Since spatial arrangements can vary widely based on context and user intent, SFT cannot cover underlying distribution of plausible layouts, restricting model's adaptability and generalization. In contrast, MetaSpatial leverages RL to overcome SFT's limitations, replacing fixed annotations with reward-driven learning and eliminating the need for post-processing.

Existing approaches often struggle with physical plausibility, coherence, and structural consistency. To address these challenges, some methods adopt multi-agent/round refinement, where LLMs refine layouts through reasoning and search during inference (Tung & Yuan, 2007). However, they are time-consuming and prone to deadlocks, where iterations fail to converge. Others leverage VLMs' multi-modal reasoning with asset and room images to improve spatial arrangement, yet they still suffer from inconsistencies and often require heavy post-processing such as differentiable optimization in LayoutVLM (Sun et al., 2024). While SFT is proposed to reduce post-processing overhead (Sun et al., 2024), its applicability to spatial reasoning is limited because layout generation lacks a definitive ground truth—there is no single "correct" layout but a distribution of valid ones. This ambiguity arises from two factors: (1) for the same room and user prompt, multiple arrangements can be equally valid (e.g., sofa by the window vs. against the wall); and (2) spatial coordinates are continuous, so small deviations that avoid collisions or violations remain acceptable. Since SFT depends on single-target annotations, it fails to capture such diversity and continuity, limiting generalizable reasoning. In contrast, RL is inherently well-suited for this task, as it learns from evaluative feedback rather than static labels, optimizing for spatial plausibility under physics constraints and layout principles. By replacing rigid supervision with adaptive learning, RL equips models with flexible spatial reasoning and enables coherent and realistic 3D scenes without post-processing.

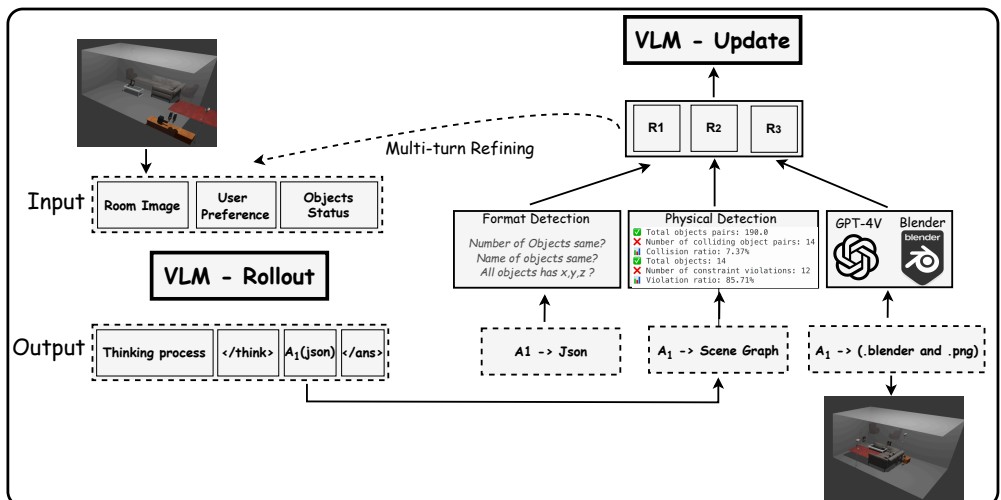

Figure 1: Overview of MetaSpatial framework. Given room images, user preferences, and object status, the model generates a JSON-formatted layout with precise (x, y, z) coordinates and a reasoning process. It evaluates the layout using three reward signals: Format Detection, Physical Detection, and Rendering-based Evaluation. The RL updates are based on multiple multi-turn refinement trajectories, optimizing a grouped policy via our 3D-SPO to learn deeper spatial reasoning.

To this end, we propose MetaSpatial, an RL-based training framework that equips VLMs with generalizable 3D spatial reasoning for scene layout generation. MetaSpatial directly optimizes spatial structures through interaction-driven rewards, enabling models to internalize physics constraints and layout principles without relying on annotations or heavy post-processing. This yields more coherent, efficient, and realistic layouts. Empirically, MetaSpatial improves both spatial plausibility and overall layout quality, highlighting RL as a promising paradigm for scalable, real-time 3D scene generation.

As shown in Figure 1, given a room image, a user preference, and the existing objects' status, the VLM generates a reasoning trace with a JSON-formatted layout specifying object positions in (x, y, z) coordinates. This layout is evaluated by three validation mechanisms to provide reward signals for RL optimization: (1) **Format Detection** verifies structural validity by checking predicted objects counts, ID consistency, and coordinate completeness; (2) **Physical Detection** converts the layout into a scene graph to assess spatial constraints, collisions, and physical rules violations; and (3) **Rendering-based Evaluation** renders a scene with predicted object coordinates and use a powerful VLM (GPT-4o) to score plausibility, aesthetic coherence, and preference alignment. MetaSpatial further employs a multi-turn training-only refinement pipeline, where each turn refines a layout based on prior experiences to form an improvements trajectory. Rather than updating the policy based on only trajectory-level rewards (Shao et al., 2024), we propose the **3D-SPO** to integrate object-level physics-informed modulation for 3D spatial reasoning. Specifically, 3D-SPO applies mask-aware penalty adjustments to advantage estimates, targeting tokens for (x, y, z) coordinates. This allows the RL to prioritize learning from physical spatial feedback rather than treating all tokens equally. By grouping trajectories, 3D-SPO further stabilizes training and ensures advantage estimates capture object-level spatial consistency and global reward trends, fostering deeper spatial reasoning.

In summary, our work makes the following key contributions:

- We introduce **MetaSpatial**, the first RL-based framework for enhancing 3D spatial reasoning in VLMs, enabling coherent 3D scene layout generation without extensive post-processing.
- We propose **3D-SPO**, a novel training schema that integrates object-level physics-aware modulation and trajectory-level reward aggregation from a training-only multi-turn refinement pipeline, enhancing temporal credit assignment and spatially consistent policy learning.
- We design a three-level **evaluation mechanism**, incorporating format detection, physical detection, and rendering-based assessment, providing adaptive reward signals for RL.
- Our extensive experiments show the effectiveness of our approach and the improvements in spatial coherence, physical plausibility, and scene quality.

## 2 METHODOLOGY

In this section, we first formalize the task of 3D scene layout generation. We then provide an overview of the MetaSpatial, highlighting its key components. Later, we detail each module within the RL training pipeline, including layout generation via VLMs, the construction of multi-turn refinement trajectories, reward design, and our core contribution—3D Spatial Policy Optimization (3D-SPO), which leverages trajectory-level feedback to guide spatial policy learning.

### 2.1 PROBLEM FORMULATION

We aims to 3D scene layout generation to place a set of given 3D objects within a specified room according to spatial constraints and user preferences. Formally, **for each single data**, given an input consisting of a room image $r$, a list of object candidates $O = \{o_1, o_2, \ldots, o_n\}$ (each annotated with category, size, and material), and optional user instructions $u$, the model is required to generate a 3D layout $l = \{(o_i, x_i, y_i, z_i)\}_{i=1}^{n}$, where each object is assigned a precise position $(x, y, z)$.

This task is inherently ill-posed: for the same input, multiple valid layouts can satisfy both physical and semantic constraints. As such, traditional SFT that relies on fixed annotations fails to capture the diversity and adaptability required for realistic spatial reasoning. Instead, we model layout generation as a policy learning problem, where a vision-language model $\pi_\theta$ is optimized to generate semantically meaningful and physically consistent layouts through interaction with a spatial feedback environment.

### 2.2 OVERVIEW OF THE METASPATIAL FRAMEWORK

As illustrated in Figure 1, **MetaSpatial** is an RL-based framework designed to enhance 3D spatial reasoning in VLMs. The core idea is to enable VLMs to not only generate initial scene layouts via inference but also iteratively refine them through multi-turn trajectories guided by structured reward signals. Given a room image, a user preference, and objects metadata, the model produces both a reasoning trace and a JSON-formatted layout prediction specifying precise $(x, y, z)$ coordinates for each object. The predicted layout is evaluated using three complementary reward functions: (1) **Format Detection**, which verifies structural correctness such as object count, ID consistency, and coordinate validity; (2) **Physical Detection**, which assesses physical feasibility by detecting object collisions, boundary violations, and other physical constraints; and (3) **Rendering-based Reward**, which renders the predicted layout and uses powerful VLM to score scene realism, aesthetic quality, and alignment with user intent. Unlike conventional RL frameworks that rely on single-turn feedback, MetaSpatial adopts a multi-turn training-only refinement pipeline, then generates multiple multi-refinement trajectories of layout updates per sample. These trajectories are grouped and optimized using our proposed **3D-SPO**, which integrates object-level physics-informed modulation to prioritize learning on coordinate tokens and trajectory-level reward aggregation to stabilize policy optimization.

### 2.3 LAYOUT GENERATION WITH VISION-LANGUAGE MODELS

At the core of our framework is a VLM $\pi_\theta$ responsible for generating a structured 3D layout given the input context. Specifically, **for each single data**, the model receives a visual prompt composed of: (1) a rendered room image $r$ that provides global spatial context; (2) a list of object specifications $O$ describing the target objects' names, categories, sizes, and styles; and (3) optional user preferences $u$ in natural language (e.g., "place the dining table in the center of the room"). The VLM processes this multimodal input and generates a roll-out $rol$ combined with two components: (1) A reasoning trace that reflects the model's spatial thinking process in natural language, outlining the logic behind each object placement; and (2) A JSON-formatted layout that encodes the final predicted positions for each object. Then, our multi-turn training-only refinement pipeline evaluates and refines the layout to generate a trajectory. Worth noting, the initial input image is a rendered empty room that serves only as a blank spatial canvas, not as a supervision signal; in later turns, the generated scene is rendered and fed back as visual context, enabling iterative refinement.

### 2.4 MULTI-TURN REFINEMENT TRAJECTORY GENERATION

Unlike traditional single-turn optimization, MetaSpatial introduces a multi-turn **training-only** refinement pipeline that enables the model to iteratively improve layouts by reflecting on previous turns.

The motivation is distinct from prior multi-turn refinement methods applied during inference, although such methods can still be optionally used after training. Formally, instead of generating a single-step roll-out $rol$ **per sample**, we produce a T-turn refinement trajectory $\mathcal{T} = \{rol_1, rol_2, \ldots, rol_T\}$, where $rol_1$ is the initial layout and subsequent roll-outs $rol_t$ $(t > 1)$ update object placements conditioned on the previous roll-out $rol_{t-1}$ and its environment feedback. At each turn, the model re-generates both a reasoning trace and a layout proposal, forming a sequence of progressively refined outputs. To guide learning, we apply a discounted cumulative reward across the trajectory, assigning higher weight to earlier turns. This encourages the model to produce a strong initial layout while still benefiting from iterative refinement during training. These trajectories serve two purposes: (1) They expose the model to diverse layout revisions, promoting structural adaptability and robust spatial understanding; (2) They allow reward comparison not just across samples, but *within* refinement paths, which is essential for the 3D-SPO strategy introduced later; and (3) They accelerate training by providing multiple learning signals per sample, enabling faster convergence with fewer optimization steps and reduced wall-clock time, demonstrated in experiments. This refinement-based approach aligns with how humans iteratively adjust object arrangements in space and provides the necessary supervision signal for layout optimization in the absence of absolute ground truth.

## 2.5 REWARD DESIGN

To guide RL without explicit annotations, MetaSpatial adopts a hybrid reward mechanism with three complementary components—format, physical, and rendering—that jointly capture layout quality and enable fine-grained spatial reasoning. To ensure efficient and stable training, we applied staged tuning: format reward was emphasized early to address basic instruction-following (e.g., object count, naming), physical reward was increased once format accuracy exceeded 0.9 to enhance spatial reasoning, and rendering reward was introduced only in later stages due to its high runtime, refining visual plausibility without slowing early training. We first extract a predicted layout $l_t$ from $rol_t$ at step $t$, we define the total reward $R(l_t)$ as a weighted sum:

$$R(l_t) = \lambda_1 R_{\text{format}} + \lambda_2 R_{\text{physics}} + \lambda_3 R_{\text{render}} \tag{1}$$

**(1) Format Reward.** This component assesses whether the generated output adheres to the expected structural format, including both syntactic and semantic validity. Instead of assigning a binary score, we use a graded reward function that evaluates the following aspects: (a) *Tag Structure Check:* If output mismatches the expected pattern: a reasoning trace enclosed in `<think>` tags and a corresponding layout enclosed in `<answer>` tags, **0**. (b) *JSON Parsing Check:* If the layout section cannot be parsed into a valid JSON object, **0.1**. (c) *Object Count Consistency:* If the number of predicted objects mismatches the given objects, **0.5**. (d) *Name Alignment Check:* If the predicted object IDs misalign with the given objects, **0.5**. (e) *Coordinate Validity Check:* If any generated objects do not have valid `x`, `y`, and `z` coordinates, **0.5**. (f) *Full Match:* If all checks pass, **1**.

Formally, the format reward $R_{\text{format}} \in \{0, 0.1, 0.5, 1.0\}$ is determined by parsing the model output and applying a sequence of rule-based validators on both structure and content. This design allows the model to receive meaningful gradients even for partially correct outputs, facilitating more stable learning during early training stages.

**(2) Physics Reward.** To ensure that the generated layout adheres to fundamental physical constraints, we simulate spatial arrangements by converting the predicted JSON layout into a scene graph and performing rule-based physical checks. Two major types of violations are penalized: (a) *Collision Ratio:* Measures the proportion of objects that intersect or overlap in space, computed via bounding box intersection in 3D space. (b) *Constraint Violation Ratio:* Measures how often objects are placed outside of allowable spatial bounds, such as floating in midair or extending beyond room boundaries.

The physics reward is computed as:

$$R_{\text{physics}} = -\alpha \cdot \text{CollisionRatio} - \beta \cdot \text{ConstraintRatio} \tag{2}$$

where $\alpha$ and $\beta$ are weight factors (set to 0.2 by default). This component promotes physically valid layouts and discourages unrealistic object placements.

**(3) Rendering-based Reward.** To assess the overall realism, functionality, and aesthetic coherence of a generated scene, we adopt a rendering-based evaluation strategy using GPT-4o as a vision-language judge. Specifically, the predicted layout is rendered into a 3D scene image using Blender, and

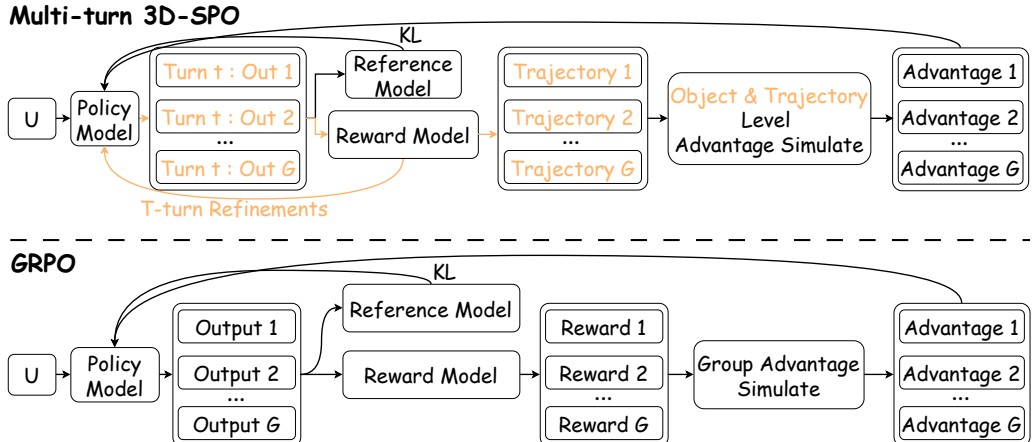

Figure 2: Comparison between **Multi-turn 3D-SPO** framework and standard GRPO. As highlighted by the orange components, 3D-SPO introduces a multi-turn refinement pipeline that transforms each single-step output in GRPO into a $T$-step trajectory with structured rewards. These trajectories are aggregated and processed by our proposed *dual-level advantage simulator*, which embeds physics-informed spatial penalties and produces advantage estimates at both the object and trajectory levels.

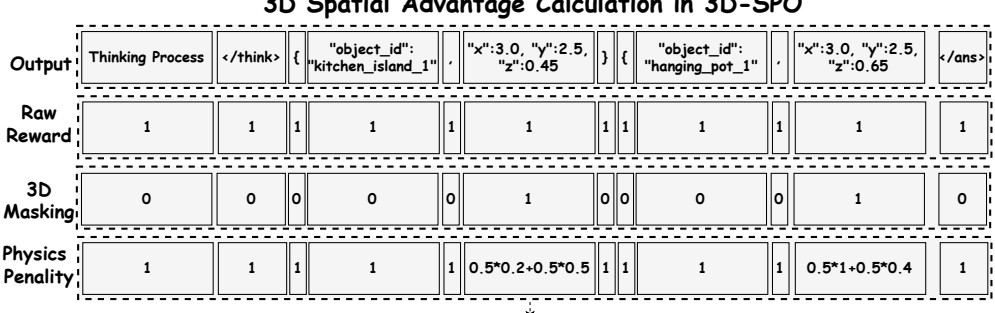

Figure 3: Advantage Calculation in 3D-SPO. Special weights are applied to tokens corresponding to 3D coordinates. First, a 3D masking mechanism identifies all tokens representing the x, y, and z positions of objects. For each object, a physics penalty is computed based on the collision ratio and constraint ratio derived from the reward signal. These penalties are weighted and multiplied by the original reward. All adjusted rewards are then normalized by subtracting the trajectory-level group average and dividing by the standard deviation to deliver the final advantages.

the image is sent to GPT-4o along with the user's textual preferences as shown in Appendix F.1. The model is prompted to rate the layout across five human-aligned criteria: (1) Realism and 3D Geometric Consistency; (2) Functionality and Activity-based Alignment; (3) Layout and Furniture Appropriateness; (4) Color Scheme and Material Choices (we fixed this score due to the fixed objects in setting); and (5) Overall Aesthetic and Atmosphere. Each category is graded on a scale from 1 to 10, and the final reward is computed as the normalized sum: $R_{\text{render}} = \frac{1}{50} \sum_{i=1}^{5} \text{Grade}_i$.

This high-level reward signal captures subjective qualities that are hard to model directly, such as stylistic alignment and visual appeal, and serves as a proxy for human feedback in training.

## 2.6 MULTI-TURN 3D SPATIAL POLICY OPTIMIZATION (3D-SPO)

Inspired by GRPO (Shao et al., 2024), we propose the **Multi-Turn 3D Spatial Policy Optimization (3D-SPO)** algorithm to optimize the policy model $\pi_\theta$. For each training sample, we parallelly collect multiple refinement trajectories with their associated discounted rewards. 3D-SPO leverages these grouped trajectories for relative reward comparisons, incorporates both trajectory-level and object-level feedback, and focuses policy updates on generated 3D coordinates, thereby enhancing spatial reasoning in both local and global aspects.

In detail, we generates $G$ parallel refinement trajectories $\zeta = \mathcal{T}_1, \mathcal{T}_2, \ldots, \mathcal{T}_G$ **per input sample**, where each trajectory is defined as $\mathcal{T}_g = \{rol_{g,1}, \ldots, rol_{g,T}\}$, consisting of a sequence of layout versions. Each layout $l_{g,t}$ extracted from $rol_{g,t}$ is assigned a composite reward $R(l_{g,t})$ based on three factors: format validity, physical plausibility, and aesthetic quality. Unlike prior work that uses last-turn reward as final trajectory reward, we use a discounted cumulative reward as $R_g = \sum_{i=1}^{T}\{\gamma^t \cdot R(l_{g,t})\}$, where $\gamma \in (0, 1)$ is a decay factor that places greater emphasis on early layout quality. This design ensures that earlier turns contribute more to the final reward than later ones, encouraging the model to produce high-quality layouts as early as possible rather than relying on prolonged optimization. This mechanism aims that longer sequences are not inherently favored, as late rewards are down-weighted.

As shown in Figure 3, we estimate the baseline from group scores instead of using the value model, inspired by GRPO. In the advantage estimate, we first utilize a 3D masking mechanism to identify the coordination tokens of all objects. For each object, we can get a physics penalty by the collision ratio and constraint ratio derived from previous physical detection reward. Then, for coordinate tokens, these penalties are weighted and multiplied by the original reward to provide a new trajectory-level reward $\hat{R}_g$. Non-coordinate tokens retain the original ones. This allows the model to focus more on spatially important tokens (coordinates) while preserving trajectory- level consistency for other tokens. After adjustments, we normalize them by subtracting the original group average reward $\mu$ and dividing by the original standard deviation $\sigma$ across the G trajectories. In our setting, the normalized reward is used directly as the advantage for all tokens in i-th trajectory as $\hat{A}_{i,k}^{3D} = (\hat{R}_{i,k} - \mu)/\sigma$.

After the advantage estimate, we optimize the VLM policy $\pi_\theta$ using our designed objective. We adapt the original GRPO objective by incorporating our physics-aware advantage estimate $\hat{A}_{i,k}^{3D}$ as follows:

$$\mathcal{J}_{3D\text{-}SPO}(\theta) = \mathbb{E}[q \sim P(Q), \{\mathcal{T}_i\}_{i=1}^{G} \sim \pi_{\theta_{old}}(\zeta|q)]$$

$$\frac{1}{G}\sum_{i=1}^{G}\frac{1}{|\mathcal{T}_i|}\sum_{k=1}^{|\mathcal{T}_i|}\left\{\min\left[rto_{(i,k)}\hat{A}_{i,k}^{3D}, \text{clip}\left(rto_{(i,k)}, 1-\epsilon, 1+\epsilon\right)\hat{A}_{i,k}^{3D}\right] - \beta\mathbb{D}_{KL}\left[\pi_\theta||\pi_{ref}\right]\right\},$$

$$\tag{3}$$

where $rto_{(i,k)} = \frac{\pi_\theta(\mathcal{T}_{i,k}|q,\mathcal{T}_{i,<k})}{\pi_{\theta_{old}}(\mathcal{T}_{i,k}|q,\mathcal{T}_{i,<k})}$ is the likelihood ratio, and $\hat{A}_{i,k}^{3D}$ is our 3D-SPO advantage on i-th trajectory and k-th token, $\pi_{old}$ is the behavior policy used to generate rollouts, $\pi_{ref}$ is the frozen reference policy, and $\mathcal{D}_{KL}$ is the KL divergence term to prevent over-exploration.

This formulation offers three key advantages: (1) It integrates object-level physics-aware modulation in advantage estimate, allowing the model to focus learning on coordinate-relevant tokens and better internalize spatial constraints; (2) It encourages the generation of high-quality layouts in early refinement steps through discounted reward accumulation, providing trajectory-level supervision without requiring ground-truth annotations; and (3) It supports stable, group-wise policy updates that improve generalization to diverse and complex scene configurations.

## 3 EXPERIMENTS

Here, we evaluate MetaSpatial's effectiveness in improving spatial reasoning and 3D layout generation capabilities of VLMs through reinforcement learning. We present experimental setups, quantitative metrics, qualitative comparisons, and ablation analyses to demonstrate the benefits of MetaSpatial. In addition, to demonstrate the zero-shot generalization improvements, we evaluate models on the Open3DVQA (Zhan et al., 2025) benchmark as shown in Appendix G.

### 3.1 EXPERIMENTAL SETUP

We conduct experiments using Qwen2.5-VL 3B and 7B as our base VLMs. All models are trained on a curated dataset of indoor 3D scenes, where each scene includes a room image, a list of objects with specifications, and a user preference as shown in Appendix D. **Notably**, our dataset does not contain ground-truth object coordinates; it only provides textual room descriptions and a corresponding asset library. Instead of relying on ground-truth layouts, MetaSpatial is trained purely through interaction and feedback using our custom reward function. We also include other scene layout generation methods: I-Design (Çelen et al., 2024), LayoutGPT (Feng et al., 2023). For rendering, we use Blender

Table 1: Performance comparison across models with and without RL. RL leads to consistent improvements in formatting accuracy, physical feasibility, and perceptual scene quality.

| Model | Format ↑ | GPT-4o Score ↑ | Collision ↓ | Constraint ↓ | Overall |
|---|---|---|---|---|---|
| Qwen 3B | 0.12 | 0.03 | 79.0% | 100% | -0.27 |
| Qwen 3B + MetaSpatial | 0.49 | 0.18 | 68.5% | 100% | -0.09 |
| Qwen 7B | 0.85 | 0.35 | 38.2% | 95.5% | 0.51 |
| Qwen 7B + MetaSpatial | **0.98** | 0.62 | **11.5%** | **70.8%** | **0.95** |
| GPT-4o | 0.95 | 0.58 | 26.3% | 79.4% | 0.87 |
| I-Design | - | **0.64** | 22.5% | 83.3% | 0.92 |
| LayoutGPT | - | 0.55 | 20.7% | 80.2% | 0.85 |

and deploy it in a dedicated server. For each generated layout, the system places 3D assets using the predicted (x, y, z) coordinates through scripted Python. Since the room geometry and asset set are predefined (but unlabeled), this enables fully automated rendering without annotations. The renderer produces high-resolution images from a fixed, human-readable angle for visual evaluation, including GPT-4o-based perceptual scoring.

Our total reward is computed as a weighted combination of four components, directly reflecting the priorities in our learning objective:

$$R(\mathcal{L}_t) = \frac{1}{50} R_{\text{render}} + 0.5 \cdot R_{\text{format}} - 0.2 \cdot \text{CollisionRatio} - 0.2 \cdot \text{ConstraintVioRatio}, \quad (4)$$

where $R_{render}$ is the aggregated GPT-4o score, obtained by evaluating rendered scene images across five criteria (Realism, Functionality, Layout, Color Scheme, and Aesthetic), each scored from 1 to 10 and normalized by 50; $R_{format}$ is a structured format reward with values in $\{0, 0.1, 0.5, 1.0\}$, based on whether the model output is correctly structured, parsable, and matches the expected number of objects; *CollisionRatio* is the percentage of objects that overlap with others in the 3D layout, penalizing physically implausible scenes; *ConstraintVioRatio* is the proportion of objects violating spatial constraints, such as exceeding room boundaries or being improperly placed.

This reward composition guides RL by promoting physically valid and aesthetically pleasing layouts while penalizing malformed or unrealistic ones. We apply multi-turn refinement and 3D-SPO to update the model with trajectory-aware gradients during training. In comparison, we only use single-turn inference to compare the final results without multi-turn refinements.

## 3.2 QUANTITATIVE RESULTS

We report results across three metrics: (1) format correctness (i.e., structurally valid JSON outputs), (2) physical feasibility (collision and constraint violation ratios), and (3) GPT-4o-assessed layout quality. Table 1 presents the performance of Qwen-VL models (3B and 7B) with and without MetaSpatial. We observe that MetaSpatial significantly improves all metrics. For format correctness, MetaSpatial enables models to better conform to structured output expectations, with accuracy rising from 0.12 to 0.49 in the 3B model and from 0.85 to 0.98 in the 7B model. In terms of physical feasibility, RL training reduces the collision rate by 10.5% for the 3B model and 26.7% for the 7B model, while also lowering the constraint violation ratio, especially in the 7B setting. Importantly, the GPT-4o-based perceptual scores—used as a proxy for overall layout realism, coherence, and alignment with user preference—show notable gains: from 0.03 to 0.18 for Qwen-VL 3B, and from 0.35 to 0.62 for Qwen-VL 7B. These improvements are reflected in the final composite scores as well, where Qwen-VL 7B with RL achieves 0.95 compared to 0.51 without RL. Overall, the results demonstrate that MetaSpatial enhances the spatial reasoning and generation quality of VLMs, and that larger models benefit more from multi-turn refinement and structured reward feedback.

Furthermore, we compare MetaSpatial-trained Qwen-VL 7B against strong baselines, including GPT-4o, LayoutGPT, and I-Design. Results show that Qwen-VL 7B, after MetaSpatial training, outperforms these closed or multi-round systems in most metrics, particularly in physical feasibility. Specifically, our model achieves lower collision and constraint violation rates, demonstrating its superior ability to internalize spatial rules and generate physically plausible layouts.

Table 2: Ablation study of reward components on Qwen2.5-VL 7B.

| Reward Setting | Format ↑ | GPT-4o Score ↑ | Collision ↓ | Constraint ↓ |
|---|---|---|---|---|
| Full Reward (Ours) | **0.98** | **0.62** | **11.5%** | **70.8%** |
| w/o Rendering ($\lambda_3 = 0$) | 0.96 | 0.45 | 14.5% | 80.5% |
| w/o Physics ($\lambda_2 = 0$) | 0.97 | 0.40 | 35.0% | 89.6% |
| w/o Format ($\lambda_1 = 0$) | 0.72 | 0.41 | 16.3% | 84.8% |

Table 3: Comparison of single-step RL and our multi-turn refinement strategy with 3D-SPO

| Method | Format ↑ | GPT-4o Score ↑ | Collision ↓ | Constraint ↓ |
|---|---|---|---|---|
| One-step RL (PPO) | 0.97 | 0.44 | 26.6% | 83.0% |
| Multi-turn RL (GRPO) w/ T = 1 | 0.96 | 0.5 | 21.3% | 81.2% |
| Multi-turn RL (3D-SPO) w/ T = 1 | 0.97 | 0.53 | 20.3% | 77.0% |
| Multi-turn RL (GRPO) w/ T = 3 | 0.96 | 0.54 | 16.0% | 79.5% |
| Multi-turn RL (3D-SPO) w/ T = 3 | 0.97 | 0.60 | 14.7% | 74.3% |
| Multi-turn RL (GRPO) w/ T = 5 | 0.98 | 0.58 | 13.7% | 76.2% |
| Multi-turn RL (3D-SPO) w/ T = 5 | 0.98 | **0.62** | **11.5%** | **70.8%** |
| Multi-turn RL (GRPO) w/ T = 7 | **0.99** | 0.55 | 15.3% | 78.5% |
| Multi-turn RL (3D-SPO) w/ T = 7 | **0.99** | 0.59 | 13.9% | 75.2% |

## 3.3 QUALITATIVE RESULTS

Figure 4 illustrates qualitative comparisons between scenes generated before and after RL training. Prior to reinforcement learning, object placements are often misaligned, physically implausible, and visually cluttered, with issues such as floating or overlapping items. After applying MetaSpatial, layouts become significantly more structured and realistic—objects are better aligned, grounded, and arranged in contextually appropriate positions. These improvements confirm that RL enables VLMs to internalize spatial constraints and generate more coherent, functional 3D scenes, demonstrating its value for real-world applications like AR/VR, metaverse design, and game development.

## 3.4 ABLATION STUDY

We further analyze the contribution of each reward component by training models with partial reward configurations. Table 2 shows that removing any reward component leads to degraded performance, especially when the rendering-based reward is omitted. Table 6 demonstrates that exposing the model's step-wise spatial rationale provides a stronger learning signal for RL. These results verify that all components contribute to robust and reliable spatial reasoning.

For **performance**, we compare the training-only multi-turn refinement strategy with a one-step RL baseline (Proximal Policy Optimization, PPO), as well as GRPO and 3D-SPO variants using different refinement depths. As shown in Table 3, multi-turn refinement improves layout quality across all evaluation metrics compared to single-step optimization. 3D-SPO method achieves the best overall performance, significantly reducing collision and constraint violation rates. Notably, 3D-SPO outperforms GRPO under the same number of refinement steps, particularly in physical plausibility metrics. For example, with $T = 5$ refinement steps, 3D-SPO achieves a collision rate of 11.5% and a constraint violation rate of 70.8%, both lower than the GRPO counterpart. These improvements highlight the benefit of incorporating object-level physics-aware modulation during advantage estimation. We also observe that increasing the number of refinement steps generally boosts performance; however, this trend does not hold indefinitely. When $T = 7$, performance slightly degrades compared to $T = 5$, suggesting that excessive refinement may lead to over-adjustments or reward saturation. This finding highlights the importance of balancing refinement depth with policy stability during training. For **training Efficiency**, while multi-turn refinement adds computation during rollout, its primary benefit is accelerating training. In early single-turn experiments, convergence was much slower—even without rendering—while format and constraint scores improved only gradually. Multi-turn refinement, by contrast, generates multiple layouts per prompt, yielding richer supervision and enabling discounted cumulative rewards that encourage high-quality layouts from the first step, reducing reliance on multi-step inference at test time. Moreover, for large models (e.g., 7B), training time is dominated by parameter updates and checkpointing rather than rollout, so multi-turn refinement increases learning signal without proportional update cost. Empirically, achieving similar performance with single-turn training required about 2× more

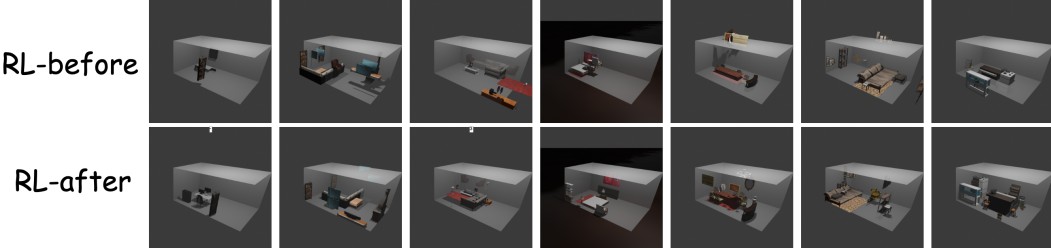

Figure 4: **Before vs After**: It highlights the efficacy of MetaSpatial in improving 3D spatial reasoning.

optimization steps and 2.5× longer wall-clock time than our multi-turn setup. For **reasoning trace influence**, including a reasoning trace yields consistent gains: GPT-4o perceptual score rises from 0.41 to 0.52, collision rate drops by 6.8 pp (34.2% → 27.4%), and constraint violations fall by 6.6 pp (87.9% → 81.3%). Format accuracy also improves slightly (0.85 → 0.87) as shown in Table 6. These results indicate that exposing the model's step-wise spatial rationale provides a stronger learning signal for RL, improving both semantic layout quality and physical plausibility. We also provide a reasoning example of trained model in Example 2.

### 3.5 SUPERVISED FINE-TUNING WITH HIGH-REWARD LAYOUTS

To further explore whether the highest-reward rollouts can serve as pseudo-annotations for SFT, we constructed an SFT dataset from layouts whose composite reward exceeded a threshold (0.8). We conducted three sets of experiments:

**SFT from High-Reward Layouts.** Using Qwen-7B as the base model, we collected 218 pseudo-SFT samples within 100 RL steps and trained a model solely on this data. As shown in Table 7, SFT notably improved *format accuracy* (0.96 vs. 0.87) but underperformed RL in terms of *spatial reasoning* and *physical feasibility* (e.g., GPT-4o score dropped from 0.52 to 0.42, and collision rate increased slightly from 27.4% to 30.5%).

**Cold-Start SFT + RL Fine-Tuning.** We then adopted the SFT-trained model as a warm-start and continued training with RL for 100 steps. This hybrid strategy yielded strong results: format accuracy reached 0.98, GPT-4o score improved to 0.60, and collision rate decreased to 13.4%. Remarkably, these results nearly match our full RL pipeline but required only half the number of RL steps, demonstrating that SFT provides a highly efficient initialization.

**Extension to Smaller Models.** Finally, we tested the cold-start strategy on a weaker 3B model that initially struggled with format accuracy. After SFT-based warm-start and 100 RL steps, the model achieved 0.88 format accuracy and surpassed our reported baseline in all metrics (GPT-4o score: 0.34; collision: 33.6%; constraint satisfaction: 81.5%).

**Summary.** These experiments show that high-reward rollouts can be leveraged as pseudo-labeled SFT data to enhance *format fidelity* and accelerate RL convergence. While pure SFT improves structural correctness, the combination of SFT cold-start and RL fine-tuning provides both efficiency and robustness, particularly benefiting smaller models.

## 4 CONCLUSION, LIMITATION, AND FUTURE WORK

We introduce MetaSpatial, the first reinforcement-learning framework that equips vision–language models with robust 3D spatial reasoning. MetaSpatial combines a three-level reward scheme (format validity, physics consistency, and rendering quality) with a multi-turn refinement pipeline optimized by an enhanced 3D-SPO algorithm. This design lets a VLM produce physically plausible, structurally coherent, and aesthetically pleasing 3D layouts—without post-processing or large annotated datasets. Experiments confirm substantial gains in layout quality and adaptability over standard supervised baselines. Looking ahead, we will develop lighter rendering and evaluation pipelines to cut computational cost, extend MetaSpatial to open-world object retrieval and multi-room scenes, and explore its transferability to domains such as robotic planning, AR/VR scene design, and embodied AI.

## 5 ACKNOWLEDGMENTS

We gratefully acknowledge support from the NVIDIA Academic Grant ("Interactive Spatial Reasoning and 3D Scene Generation with RL-Enhanced VLMs") and the provision of cloud computing resources, which enabled systematic training and evaluation of our MetaSpatial and other baselines. This paper is a core component of that project. The views expressed are those of the authors and do not necessarily reflect those of NVIDIA.

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

# Appendix

## A   BROADER IMPACTS

This work has both potential positive and negative societal impacts. On the positive side, MetaSpatial enables more realistic and physically coherent 3D scene layouts, which can benefit applications in AR/VR design, robotics, education, and interior planning (Zhang et al., 2025). The framework could make spatial reasoning more accessible and scalable, lowering the barrier for creators, architects, and simulation developers to design virtual environments or train agents in physically grounded simulations. However, the same technology may also introduce risks if misused. High-fidelity 3D layouts could be applied in disinformation contexts, such as producing synthetic environments for propaganda or manipulation in virtual spaces. Additionally, if such models are trained on biased or unrepresentative data, they may reinforce cultural stereotypes in the generated layouts (e.g., object placement norms across cultures or socioeconomic classes). To mitigate these risks, we advocate for (i) careful data auditing, (ii) controlled release of powerful models, and (iii) monitoring downstream use cases, especially in immersive or generative platforms. While our work is foundational and focuses on structured layout prediction rather than full content synthesis, we recognize its downstream impact potential and encourage responsible deployment.

## B   USE OF LLMS

In this work, large language models are used solely for the purpose of grammar correction and language polishing. All technical contributions including conceptual framework design, algorithm development, model training, experiments and paper writing are original and developed by the authors.

## C    RELATED WORK

Our work builds upon recent progress in 3D scene generation, vision-language modeling, and reinforcement learning for structured reasoning. We review three lines of related research: (1) paradigms for generating 3D scenes either through generative modeling or layout-based synthesis; (2) methods for enhancing spatial reasoning of VLMs; and (3) the emerging role of RL in enhancing the reasoning capabilities of foundation models. These perspectives contextualize the motivation and novelty of our proposed MetaSpatial framework together.

### C.1    3D SCENE GENERATION PARADIGMS

In recent years, two primary directions have emerged in 3D scene generation. The first direction leverages generative models to create 3D representations such as meshes (Schult et al., 2024; Man et al., 2024). However, the generated scenes often lack the granularity and fidelity required for downstream embodied applications, where high-quality and individually controllable objects are essential (Fang et al., 2023). With the advancement of large foundation models (Pan et al., 2024b;c;a; 2025c;b;d;e; Tang et al., 2025; Luo et al., 2025), the second direction focuses on generating interme- diate representations-namely, scene layouts-by retrieving objects from large-scale asset repositories (Çelen et al., 2024; Rahamim et al., 2024). LayoutGPT is among the first to utilize LLMs as vi- sual planners, generating layouts conditioned on text descriptions (Feng et al., 2023). However, its performance is limited by the general-purpose pretraining strategy of LLMs. I-Design further introduces a multi-agent framework, where a team of LLMs represents different roles in the design process (Çelen et al., 2024). To enhance physical plausibility, LayoutDreamer integrates 3D Gaussian Splatting to optimize the generated layout (Zhou et al., 2025). Meanwhile, LayoutVLM leverages the visual understanding capabilities of VLMs to produce enhanced representations from visually marked images, followed by a differentiable optimization process (Sun et al., 2024). This work also demonstrates that fine-tuning VLMs with existing layout data can improve generation quality. Despite these advances, two key challenges remain: (1) the lack of internalized 3D spatial reasoning in VLMs, which limits their abilities to ensure physical plausibility without costly post-processing; and (2) the inefficient of SFT, which limits generation diversity due to the absence of perfect ground truth annotations. These challenges motivate us to explore a new paradigm for layout generation.

### C.2    SPATIAL REASONING WITH VLMS

VLMs show impressive capabilities in tasks involving visual understanding and natural language generation, such as image captioning, visual question answering, and referring expression comprehen- sion (Zhang et al., 2024). However, their capacity for structured spatial reasoning—particularly in 3D environments—remains underexplored. Prior works such as BLIP-2 (Li et al., 2023) and Flamingo (Alayrac et al., 2022) exhibit limited understanding of spatial relations beyond 2D grounding or image-text alignment. Recent efforts attempt to address this by providing VLMs with more struc- tured or spatially annotated inputs such as SpatialVLM (Chen et al., 2024) and AutoSpatial (Kong et al., 2025). Yet, such approaches still depend heavily on external supervision to enforce spatial constraints. However, 3D spatial reasoning inherently lacks perfect annotations and a single ground truth—multiple valid layouts can exist for the same input, each satisfying different contextual or functional constraints. As a result, current supervised approaches struggle to capture the diversity and adaptability required for realistic scene generation. In contrast, MetaSpatial addresses this limitation by allowing VLMs to learn spatial reasoning through interactive feedback and constraint-driven exploration, moving beyond annotation-dependent paradigms.

### C.3    RL FOR ENHANCING REASONING IN FOUNDATION MODELS

RL recently re-emerges as a promising strategy for improving the reasoning capabilities of large foundation models (Guo et al., 2025). Instead of relying on traditional SFT with fixed ground truth, RL allows models to learn from evaluative feedback, which is especially useful in tasks lacking a single correct answer or exhibiting structural ambiguity (Mondillo et al., 2025). In language models, RL has been widely used in the form of Reinforcement Learning from Human Feedback (RLHF) to align model outputs with human preferences (Bai et al., 2022). Such approaches are fundamental to the success of instruction-following models such as ChatGPT (Wu et al., 2023). In addition, rule-based reinforcement fine-tuning—such as OpenAI's o1 (Jaech et al., 2024) and DeepSeek-R1 (Guo et al., 2025)—has demonstrated strong performance in mathematical reasoning (Pan et al., 2025a), code generation (Liu & Zhang, 2025), and multi-step logic tasks (Wang et al.,

2025). These methods show that verifiable rewards—such as symbolic correctness or execution-based signals—can serve as effective supervision substitutes. While RL shows strong potential in language tasks, it remains largely unexplored in multimodal or spatial contexts. Unlike these domains, 3D layout generation lacks a definitive ground truth—multiple valid solutions may exist for the same input—making supervised fine-tuning insufficient to cover the full solution space. To address this, we propose the first rule-driven RL framework for vision-language models in 3D environments, incorporating multi-turn refinement and optimization to enable adaptive, constraint-aware spatial reasoning without reliance on rigid annotations.

# D DATASET

## D.1 OVERVIEW

Our dataset is built on top of I-DESIGN (Çelen et al., 2024) and consists of **10,000** synthetic indoor scenes. We have four-stage construction pipeline: (i) prompt generation, (ii) object-list synthesis via I-DESIGN, (iii) asset retrieval from Objaverse, and (iv) layout synthesis & rendering.

## D.2 STAGE 1: ROOM–PROMPT GENERATION

- **Language model.** We employ GPT-4o with a system prompt *"You are an interior designer …"*.
- **Scale. 10 000** unique prompts.
- **Attributes.** Each prompt specifies room type, interior style, and room dimensions $(L, W, H) \in [3\,\mathrm{m}, 10\,\mathrm{m}]^2 \times [2.6\,\mathrm{m}, 4\,\mathrm{m}]$.

## D.3 STAGE 2: OBJECT-LIST SYNTHESIS VIA I-DESIGN

- Input: textual prompt from Stage 1.
- Output: an inventory of **10–20** objects, each with {category, coarse size, material}.
- Post-processing: synonym normalisation and duplicate removal.

## D.4 STAGE 3: ASSET RETRIEVAL FROM OBJAVERSE

- **Retriever.** OpenShape (Liu et al., 2023)
- **Selection rule.** Top-1 similarity, vertex count$< 100\,\mathrm{k}$, licence $\in$CC0, CC-BY.

## D.5 STAGE 4: LAYOUT SYNTHESIS AND RENDERING

- Coarse placement via I-DESIGN physics module; grid resolution $0.1\,\mathrm{m}$.
- Fine collision pass using Bullet (50 steps).
- Blender 4.2 - Cycles, $500 \times 500$ px, 35 mm camera.

Each scene yields `scene_X.jpg`, `layout_X.json`, and a folder of GLB assets.

## D.6 LIMITATIONS AND FUTURE EXTENSIONS

Current scenes contain a single room and static lighting. We plan to extend the dataset to multi-room environments with connectivity graphs, diversified lighting, and a larger long-tail object distribution.

## D.7 EXAMPLES

Figure 9 illustrates several sample scenes from our dataset and model outputs. For each case, we show the input user prompt, the rendered scene layout, and optionally a visualization of the predicted object placements.

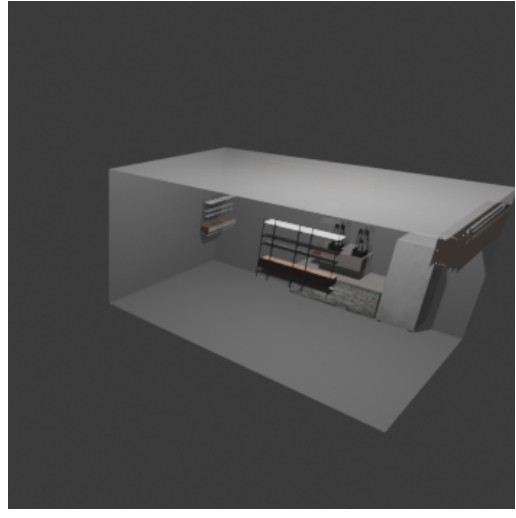

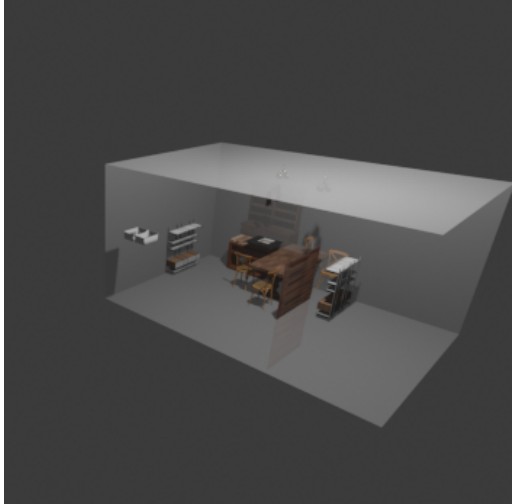

Figure 5: **Prompt:** "A bustling kitchen with pots clattering and the aroma of spices filling the air, chefs moving quickly in a culinary dance."

Figure 6: **Prompt:** "A quaint country kitchen with a farmhouse sink, checkered curtains, and a pie cooling on the windowsill."

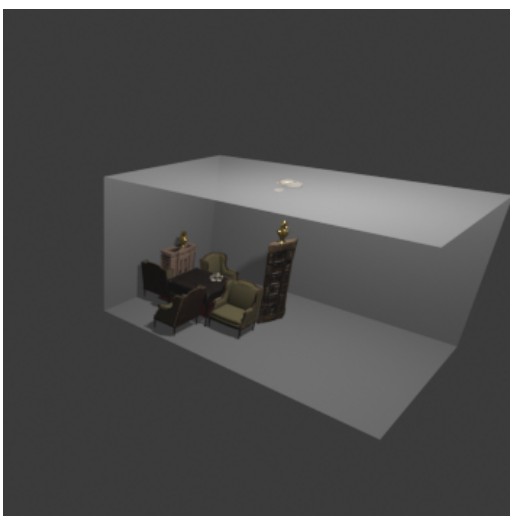

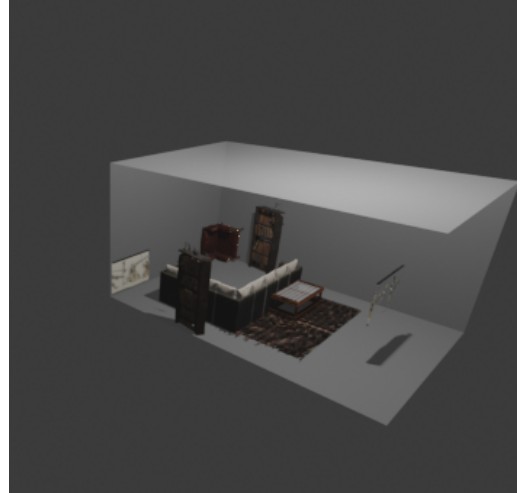

Figure 7: **Prompt:** "A small but charming tea room with a round table set with delicate china, the scent of jasmine tea lingers in the air."

Figure 8: **Prompt:** "A cozy living room with a large sectional, a coffee table stacked with books, and a plush rug underfoot."

Figure 9: Qualitative examples showing prompts, rendered layouts, and predicted object placements.

### D.8 PROMPT EXAMPLE: FULL SCENE LAYOUT GENERATION

The following is a full example of the prompt template we use to guide GPT-4o in predicting object layouts, including room description, constraints, and the JSON-encoded input. The output is expected to consist of spatial reasoning in a `<think>` block and object positions in a `<answer>` block.

Listing 1: Full Prompt Template Example for Layout Generation

```
## Task Description
You are an intelligent assistant for arranging objects in a room based
    ↪ on JSON data. The given image is the shape of the room. Your task
    ↪ is to:
1. Compute spatial coordinates for each object, respecting room layout
    ↪ and constraints.
2. Ensure logical placement by preventing object collisions and
    ↪ respecting spatial boundaries.
3. Respect user preferences when arranging objects.

## Room Information
- Room Dimensions: {'x': 6, 'y': 5, 'z': 3} (Length meter * Width meter
    ↪ * Height meter).
- Room Layout Elements (reference points for object placement):
['south_wall', 'north_wall', 'west_wall', 'east_wall', 'middle of the
    ↪ room', 'ceiling'].

## User Preferences
A bustling kitchen with pots clattering and the aroma of spices filling
    ↪ the air, chefs moving quickly in a culinary dance.

## Placement Rules:
1. Compute the Spatial Coordinates of Each Object and Generate a
    ↪ "positions" (x, y, z) field for each object.
2. Ensure Objects Do Not Collide
3. Maintain Logical Consistency

## Output Format
First reason about the placement logic and provide a step-by-step
    ↪ explanation within the <think> </think> tags. After the reasoning
    ↪ process, the final output must be structured in JSON format within
    ↪ <answer> </answer> tags. Only output the generated position of the
    ↪ objects. Do not output the original object information like style,
    ↪ material, size_in_meters, etc.

## Expected Output
<think>
Reasoning process here...
</think>
<answer>
{"new_object_id": "sectional_sofa_1","x": 1.0,"y": 1.82,"z": 0.45}
</answer>

## Input JSON Data:
[ ... full JSON omitted for brevity, see supplementary data ... ]
```

## E   REASONING EXAMPLE

Listing 2: A reasoning trace example of the trained model

```
The task requires arranging objects in a room based on given placement
    ↪ rules, ensuring spatial consistency and preventing object
    ↪ collisions. Given the input JSON data and the room dimensions and
    ↪ layout elements, I will compute the spatial coordinates for each
    ↪ object while respecting these constraints. The electric fireplace
    ↪ is placed on the south wall, but no fireplace is defined in the
    ↪ input JSON. However, I can generate a sensible position using the
    ↪ provided room dimensions and layout elements. The wall art should
    ↪ be placed on the west and east walls, and the bookshelf should be
    ↪ adjacent to the fireplace. Let's place the bookshelf on the west
```

```
    ↪ wall, using the x position determined by its size and constraints,
    ↪ and the
fireplace on the south wall. The wall art will be placed on the west and
    ↪ east walls, outside of the room dimensions to avoid collisions.
[...]
```

## F  REWARD

### F.1  RENDERING-BASED REWARD PROMPT TEMPLATE

We use the following prompt to elicit scene-level perceptual scores from GPT-4o. The model is asked to assign a grade between 1 and 10 (or `"unknown"`) across five human-aligned criteria. The prompt includes a user-defined preference and requests a structured JSON response.

Listing 3: Full Prompt Template Example for Layout Generation

```
Give a grade from 1 to 10 or unknown to the following room renders based
    ↪ on how well they correspond together to the user preference (in
    ↪ triple backquotes) in the following aspects:
    - Realism and 3D Geometric Consistency
    - Functionality and Activity-based Alignment
    - Layout and furniture
    - Color Scheme and Material Choices
    - Overall Aesthetic and Atmosphere
User Preference:
    ```{user_preference}```
Return the results in the following JSON format:
    ```json
    {example_json}
```

# G    GENERAL SPATIAL REASONING BENCHMARKS

We evaluate spatial reasoning capabilities using Open3DVQA (Zhan et al., 2025), a new benchmark with 9,000 VQA samples collected from a realistic 3D urban simulator. The dataset covers various spatial reasoning types, including relative and absolute positions, object attributes, and different viewpoints (egocentric vs. allocentric). In qualitative tasks, MetaSpatial-7B achieves the best overall accuracy (73.5%), outperforming GPT-4, GPT-4o, and both versions of Qwen-VL. Fine-tuned models like Qwen-VL-7B-SFT also show strong gains over their base versions, highlighting the benefit of targeted training. In quantitative tasks, MetaSpatial leads with the highest success rates and lowest errors, especially on challenging dimensions like vertical distance, volume, and height estimation. Compared to GPT-4 and GPT-4o, MetaSpatial shows better precision in spatial measurements. Overall, Open3DVQA reveals that (1) models reason better about relative than absolute positions and (2) RL improves performance more than SFT in 3D spatial reasoning tasks.

| Qualitative | Spatial Qualitative Relationship | | | Spatial Object Attribute | | | Overall |
| | Left/Right | Below/Above | Behind/Front | Tall/Short | Wide/Thin | Big/Small | Avg. |
|---|---|---|---|---|---|---|---|
| GPT-4 | 51.5 | 50.0 | 46.0 | 59.3 | 53.9 | 60.9 | 51.1 |
| GPT-4o | 58.5 | 49.2 | 53.9 | 58.5 | 54.6 | 68.7 | 58.7 |
| LLaVA-7B | 49.2 | 42.9 | 49.2 | 21.0 | 31.2 | 39.8 | 39.3 |
| Qwen-VL-7B | 50.0 | 48.4 | 42.9 | 52.3 | 40.6 | 42.1 | 46.9 |
| Qwen-VL-7B-SFT | 73.4 | 67.1 | 66.4 | 75.7 | 64.0 | 71.8 | 72.8 |
| **MetaSpatial-7B** | **76.9** | **72.8** | **70.3** | **78.5** | **68.2** | **74.7** | **73.5** |

Table 4: Model Performance on qualitative spatial reasoning tasks. Success rates (↑) to evaluate qualitative questions.

| Quantitative | Spatial Quantitative relationship | | | | Spatial Object Attribute | | | Overall |
| | Direct Distance | Horizontal Distance | Vertical Distance | Direction | Height | Width | Volume | Avg. |
|---|---|---|---|---|---|---|---|---|
| GPT-4 | 9.3 / 2.76 | 9.3 / 2.76 | 3.1 / >10 | 9.3 / 99.3° | 21.8 / 2.50 | 12.5 / 1.42 | 9.3 / >10 | 11.0 |
| GPT-4o | 12.5 / 0.64 | 15.6 / 1.83 | 3.1 / 11.28 | 34.3 / 66.5° | 15.6 / 0.53 | 9.3 / 0.59 | 3.1 / **0.89** | 15.3 |
| LLaVA-7B | 0.0 / 0.99 | 0.0 / 1.17 | 0.0 / 1.03 | 6.2 / 60.0° | 3.1 / 0.81 | 6.2 / 11.11 | 6.2 / 1.48 | 3.6 |
| QwenVL-7B | 3.1 / 1.69 | 3.1 / 2.83 | 3.1 / 20.87 | 18.7 / 92.8° | 3.1 / 1.67 | 6.2 / 0.87 | 0.0 / 0.94 | 5.1 |
| Qwen-VL-7B-SFT | 37.5 / 0.60 | 15.6 / 0.93 | 31.2 / 2.11 | 31.2 / 76.4° | 59.3 / 0.38 | 21.8 / 1.79 | 18.7 / >10 | 34.1 |
| **MetaSpatial-7B** | **39.2 / 0.56** | **16.5 / 0.80** | **33.7 / 0.88** | **34.6** / 75.0° | **61.4 / 0.33** | **23.7 / 0.55** | **20.9** / 0.92 | **35.6** |

Table 5: Model Performance on quantitative spatial reasoning task. We use success rates (↑) and absolute relative error (↓) to evaluate the quantitative questions.

# H    ADDITIONAL RESULTS

Table 6: Effect of reasoning traces during RL. Adding a natural-language reasoning trace improves perceptual quality and physical plausibility, with slight gains in formatting.

| Metric | RL (w/ Reasoning Trace) | RL (w/o Trace) |
|---|---|---|
| Format Accuracy ↑ | **0.87** | 0.85 |
| GPT-4o Score ↑ | **0.52** | 0.41 |
| Collision Rate ↓ | **27.4%** | 34.2% |
| Constraint Violations ↓ | **81.3%** | 87.9% |

Table 7: Comparison of RL, SFT from high-reward layouts, and hybrid strategies. Format = format accuracy; GPT-4o = GPT-4o evaluation score; ↓ = lower is better; ↑ = higher is better.

| Model / Strategy | Format | GPT-4o | Collision ↓ | Constraint ↑ |
|---|---|---|---|---|
| Qwen-7B (RL) | 0.87 | 0.52 | 27.4% | 81.3% |
| Qwen-7B (SFT) | 0.96 | 0.42 | 30.5% | 86.9% |
| Qwen-7B (SFT + RL) | 0.98 | 0.60 | 13.4% | 74.5% |
| Qwen-3B (SFT + RL) | 0.88 | 0.34 | 33.6% | 81.5% |

