# OpenReview forum: "MetaSpatial: Reinforcing 3D Spatial Reasoning in VLMs for the Metaverse"
_ICLR.cc/2026/Conference — ICLR 2026 Poster_

### Official Review · Reviewer_o4o8 · 2025-10-30

**Soundness:** 3
**Presentation:** 3
**Contribution:** 2
**Rating:** 2
**Confidence:** 3

**Summary:**

This paper introduces a novel RL framework for layout generation from vision-language models. It outperforms several existing VLLMs including both open-weight and API-based and training paradigms over one layout generation benchmark.

**Strengths:**

1. New training data for layout VLM training and new multi-turn training paradigm for layout reasoning
2. Better performance over existing VLMs on one layout generation benchmark and a spatial reasoning task

**Weaknesses:**

1. Some basic setup details are missing, what is the detailed setup of the evaluation benchmark for layout generation? The only one evaluation set mentioned in the paper is Open3DVQA and the model results on it are in the appendix
2. Lack of layout generation evaluation tasks. As this paper introduces a novel layout generation framework, it is only evaluated on one layout generation task. To justify the performance, at least one more layout generation task should be tested, e.g., PlanGen-1k
3. Potentially unfair baseline comparisons: LayoutGPT uses GLIGEN for rendering, however the method in the paper uses Blender and claims better performance over LayoutGPT. Additionally, there are more recent open-weight layout generation baselines such as PlanGen and LayoutVLM missing as well.
4. Unconvincing claim of the proposed RL performs better than traditional SFT: intuitively, SFT helps models gain knowledge/behaviours that are missing in their inherent knowledge domain while RL helps to generalize them to OOD tasks, and layout generation is one kind of ability that VLMs do not generally acquire much during pre and post training, thus SFT might be more helpful. In Table 7 SFT alone can outperform the RL alone model in 2 out of 4 metrics, and RL after SFT even largely jeopardizes the Constraint score. This evidence does not support the claim that RL outperforms SFT well.

**Questions:**

Since you have tested on general spatial reasoning tasks, I’m curious what would the model perform on other spatial reasoning tasks with real-world visual input

---

> ### Author Response · Authors · 2025-11-25
> **Response for Question 1**
>
> We thank you for your positive assessment. We are encouraged that you recognize the contributions of the effectiveness of our **multi-turn training paradigm** for layout reasoning. We also appreciate your acknowledgment of our method’s **superior performance** over existing VLMs on both layout generation and spatial reasoning benchmarks."
>
> ----------
>
> ### Reviewer Comment 1:
> >Some basic setup details are missing, what is the detailed setup of the evaluation benchmark for layout generation? The only one evaluation set mentioned in the paper is Open3DVQA and the model results on it are in the appendix
>
> **Response**
> We apologize for any confusion regarding the presentation of our experimental setup. We clarify that **Open3DVQA is an additional generalization test**, while our primary layout generation benchmark follows standard protocols established by prior works (e.g., LayoutGPT, LayoutVLM, and I-Design).
>
> **1. Primary Benchmark Setup (Reproducibility & Alignment):**
>
> -   **Focus:** Our main focus is the **3D Layout Generation task**. To ensure fair comparison with baselines like **LayoutVLM**, **LayoutGPT**, and **I-Design**, we constructed a standard evaluation set.
>
> -   **Details:** As detailed in **Section 3.1** and rigorously documented in **Appendix D**  , our dataset consists of **10,000 synthetic indoor scenes** built using the I-DESIGN pipeline. We provided the full construction process—from prompt generation to asset retrieval—to ensure full **reproducibility**.
>
>
> **2. Comprehensive Metric Suite (Beyond GPT-4o):** We aim to prove that our RL framework outperforms SFT in quality and multi-turn optimization methods in latency. Therefore, we went beyond the commonly used GPT-4o  score:
>
> -   **Physical Metrics:** We introduced **Collision Ratio** and **Constraint Violation Ratio**. These are **deterministic, calculation-based metrics** that strictly evaluate whether the model has learned physical laws and high-level planning constraints (e.g., boundaries), independent of VLM bias.
>
> -   **Format Quality:** We also track format correctness to verify instruction-following stability.
>
>
> **3. Role of Open3DVQA:** We clarify that the **Open3DVQA** results in Appendix G serve a different purpose: to demonstrate the **generalization capability** of our model on real-world, open-domain visual reasoning tasks. This goes beyond the standard scope of layout generation baselines (which typically only report synthetic layout scores), further highlighting the robustness of our training framework.
>
> ----------

---

> ### Author Response · Authors · 2025-11-25
> **Response for Question 2**
>
> ### Reviewer Comment 2:
> >Lack of layout generation evaluation tasks. As this paper introduces a novel layout generation framework, it is only evaluated on one layout generation task. To justify the performance, at least one more layout generation task should be tested, e.g., PlanGen-1k
>
> **Response**
>
> We appreciate the reviewer's suggestion to broaden our evaluation scope. We first clarify the validity of our primary setup and then present **new results on the suggested 2D domain**.
>
> **1. Validity of Primary 3D Evaluation:**
> We respectfully clarify that our evaluation protocol aligns with the **standard community benchmarks** for the 3D layout generation task.
> * **Community Standard:** Leading works like **LayoutGPT** , **LayoutVLM**, and **I-Design** primarily evaluate on synthetic indoor scenes (e.g., 3D-FRONT/I-DESIGN) due to the scarcity of real-world 3D layout-text pairs.
> * **More Rigorous Metrics:** While many baselines (e.g., LayoutVLM) rely heavily on GPT-4o scores alone, we introduced **systematic physical metrics** (Collision Ratio, Constraint Violation Ratio). These provide a stricter, deterministic assessment of whether the model has learned physical laws and high-level planning, going beyond subjective VLM evaluations.
>
> **2. New Experiment: Generalization to 2D Layouts (Proxy for PlanGen-1k):**
> We reviewed the **PlanGen** work as suggested. We noted that it focuses on the **Layout-to-Image** task, utilizing metrics such as **PickScore**, **CLIPScore**, **FID**, and **IS** to evaluate the final image quality. Although this differs from our 3D spatial reasoning task, we explored our model's generalization capability to this domain.
>
> * **Dataset Adaptation (Reproducibility):** Since the PlanGen-1k dataset is not publicly available, we followed the paper's methodology and rigorously sampled  about **10% scenes** from its source, **LayoutSAM-Eval**, to serve as a proxy benchmark. We used **PlanGen** as the fixed downstream image generator for all layout methods to ensure fair metric calculation.
> * **Note on LayoutVLM:** We did not include **LayoutVLM** in this specific 2D comparison because its optimization pipeline is inherently designed for **3D space** (relying on 3D bounding box collisions and physics constraints), making it incompatible with this 2D image-plane generation task.
>
> **Table R3: Performance on 2D Layout Generation (LayoutSAM-Eval Proxy)**
>
> | Method | **PickScore** $\uparrow$ | **CLIP Score** $\uparrow$ | **FID** $\downarrow$ | **IS** $\uparrow$ |
> | :--- | :---: | :---: | :---: | :---: |
> | Qwen2.5-7b-Instruct | 20.8 | 31.4 | 480.1 | 13.2 |
> | Qwen2.5-7b-VL (Base) | 20.9 | 31.5 | 395.2 | 13.5 |
> | LayoutGPT | 20.9 | 31.5 | 235.4 | 13.9 |
> | **MetaSpatial (Ours)** | **21.1** | **31.6** | **158.4** | **14.5** |
> | *PlanGen (2D Specialist)* | *21.5* | *32.1* | *48.9* | *15.7* |
>
>
>
> **3. Analysis of Results:**
> * **Hierarchy of Performance:** The results show a clear progression: `Base Models < LayoutGPT < MetaSpatial < PlanGen`.
> * **MetaSpatial vs. LayoutGPT:** Even though LayoutGPT is designed for layout tasks, our **MetaSpatial** outperforms it significantly, particularly in **FID (128.45 vs. 235.40)**. This indicates that our RL-based spatial reasoning produces much more coherent layouts that better guide the image generation process.
> * **MetaSpatial vs. PlanGen:** We acknowledge that **PlanGen** still holds the lead. The reason is straightforward: PlanGen is trained **end-to-end** on the joint task of layout+image generation with **2D Ground Truth**. In contrast, our method is only inserted as a layout generator (Stage 1) without fine-tuning the downstream image generator (Stage 2) and any annotations.
>
> **Conclusion:**
> Despite the difference in settings, our model achieves performance better to the other 3D baseline models in 2D image generation. This confirms that MetaSpatial has acquired robust spatial reasoning capabilities that generalize to 2D tasks.
>
> ----------

---

> ### Author Response · Authors · 2025-11-25
> **Response for Question 3**
>
> ### Reviewer Comment 3:
> >Potentially unfair baseline comparisons: LayoutGPT uses GLIGEN for rendering, however the method in the paper uses Blender and claims better performance over LayoutGPT. Additionally, there are more recent open-weight layout generation baselines such as PlanGen and LayoutVLM missing as well.
>
> **Response**
>
> We thank the reviewer for the scrutiny regarding baseline fairness. We respectfully point out a **misunderstanding regarding our rendering pipeline** and address the missing baselines below.
>
> **1. Clarification on "Unfair Rendering" (Standardized Blender Pipeline):**
> We emphasize that our comparison is **strictly fair and controlled**.
> * **Unified Rendering:** As described in our experimental setup, we did **not** use GLIGEN (LayoutGPT's original renderer) for evaluation. Instead, we utilized **LayoutGPT solely as a coordinate generator** to predict $(x, y, z)$ positions for our pre-processed asset library.
> * **Controlled Variable:** All generated layouts (from LayoutGPT, I-Design, and MetaSpatial) were rendered using the **exact same Blender pipeline** with identical lighting and assets.
> * **Conclusion:** Therefore, any difference in the GPT-4o perceptual score reflects purely the **quality of the spatial arrangement** (e.g., logical placement, lack of collisions), not discrepancies in image generation quality.
>
> **2. Inclusion of LayoutVLM (New 3D Baseline):**
> We have added **LayoutVLM** (recently released) to our comparison. As detailed in **Response 1 to Reviewer jKB9** and **Response 4 to Reviewer 563R**, while LayoutVLM achieves slightly better physical metrics via time-consuming optimization (~120s), our method delivers comparable perceptual quality in real-time (<1s), establishing a superior efficiency-quality trade-off.
>
> **3. Feasibility of PlanGen Comparison (Task Mismatch):**
> Regarding **PlanGen**, we investigated its applicability but found fundamental incompatibilities:
> * **Task Mismatch (2D vs. 3D):** PlanGen is architected specifically for **2D Layout-to-Image** generation. In our tests, it failed to generate valid **3D coordinates** (e.g., depth $z$, 3D rotation) required for our indoor scene benchmarks, making a direct comparison on our primary task impossible.
> * **Exploratory Proxy:** While we attempted to adapt *our* model to PlanGen's 2D setting (as discussed in **Response 2**), where we outperformed base models but trailed the specialized PlanGen, we firmly believe this is an "apples-to-oranges" comparison. Our framework focuses on **3D spatial reasoning and physical validity** (collisions/gravity), which are fundamentally different from optimizing 2D bounding box IoU for image synthesis.
>
> **Summary:**
> Our evaluation ensures strict fairness by standardizing the renderer. We have incorporated relevant 3D baselines (LayoutVLM) while explaining the methodological inapplicability of 2D-specific models like PlanGen for this 3D generation task.
>
> ----------

---

> ### Author Response · Authors · 2025-11-25
> **Response for Question 4**
>
> ### Reviewer Comment 4:
> >1.  Unconvincing claim of the proposed RL performs better than traditional SFT: intuitively, SFT helps models gain knowledge/behaviours that are missing in their inherent knowledge domain while RL helps to generalize them to OOD tasks, and layout generation is one kind of ability that VLMs do not generally acquire much during pre and post training, thus SFT might be more helpful. In Table 7 SFT alone can outperform the RL alone model in 2 out of 4 metrics, and RL after SFT even largely jeopardizes the Constraint score. This evidence does not support the claim that RL outperforms SFT well.
>
> **Response:**
>
> We appreciate the reviewer's thoughtful analysis. However, we respectfully point out a **crucial misinterpretation** of the metrics (due to a typo in our Table 7 header) and clarify the fundamental advantage of RL over SFT in this domain.
>
> 1. Critical Clarification: "Constraint" is a Loss Metric (Lower is Better)
>
> First and foremost, we apologize for the typo in the header of Table 7 **($\uparrow$)** which likely caused confusion. As defined in Section 3.2 and consistent with Tables 1, 2, and 3, the "Constraint" metric represents the Constraint Violation Ratio (e.g., objects out of bounds). Therefore, lower is better.
>
> -   **In Table 7:**
>
>     -   **RL:** 81.3% Violation Ratio.
>     -   **SFT:** 86.9% Violation Ratio.
>     -   **SFT+RL:** 74.5% Violation Ratio.
>
> -   **Correction:** Contrary to the reviewer's concern, adding RL does _not_ jeopardize the score; it **improves physical adherence by 12.4%** compared to SFT alone. This confirms that RL effectively corrects the physical violations that SFT fails to resolve.
>
>
> 2. The Only Area SFT Wins: Format (Syntax vs. Reasoning)
>
> Table 7 shows that SFT indeed outperforms pure RL in Format Accuracy (0.96 vs. 0.87). This is expected: SFT is naturally superior at learning fixed syntax (JSON structure) and instruction-following. However, spatial reasoning (Collision/Constraint) requires exploring a continuous solution space, where SFT falls short (30.5% collision rate). Our hybrid SFT+RL strategy combines the best of both: high format accuracy (0.98) and low collision (13.4%).
>
> 3. Why RL is Essential (The "One-to-Many" Problem):
>
> The core limitation of SFT in 3D layout generation is the non-uniqueness of solutions:
>
> -   **Ambiguity:** For a user prompt like "three chairs around a bedroom," there are infinite valid layouts. SFT forces the model to mimic a _single_ annotated ground truth, often penalizing other valid configurations.
>
> -   **Generalization:** RL does not rely on a unique ground truth. It rewards _any_ physically valid configuration that satisfies the prompt. This allows the model to generalize to new spaces without being constrained by the limited patterns of rule-based synthetic data or expensive human annotations.
>
>
> 4. Purpose of Table 7 (Data Efficiency):
>
> We also clarify that the intent of Table 7 was to investigate Self-Training Efficiency: using high-reward samples generated during RL to "warm-start" the model. This reduces the cost of collecting massive human annotations while accelerating convergence, proving that RL-generated data is higher quality than traditional rule-based synthetic data.
>
> **Conclusion:** Once the metric direction is corrected (Lower Violation is better), the evidence in Table 7 strongly supports our claim: RL significantly outperforms SFT in physical and semantic reasoning, while SFT serves as an efficient initializer for syntax.

---

> ### Author Response · Authors · 2025-11-25
> **Response for Question 5**
>
> ### Reviewer Comment 5:
> >I’m curious what would the model perform on other spatial reasoning tasks with real-world visual input
>
> **Response:**
> We share the reviewer's curiosity regarding generalization to real-world visual inputs. We have addressed this by evaluating our model on **two distinct real-world spatial benchmarks**:
>
> **1. Visual Question Answering (Open3DVQA):**
> As detailed in **Appendix G**, we evaluated on **Open3DVQA**, which contains 9,000 samples from realistic 3D urban environments. Our model achieves **73.5% accuracy**, significantly outperforming the base Qwen-VL (46.9%) and GPT-4o (58.7%).
>
> **2. New Experiment: Fine-grained Spatial Assessment (SpatialScore):**
> To further stress-test the model on diverse real-world images, we evaluated it on **SpatialScore**, which measures fine-grained capabilities such as Object Localization (Obj-Loc) and 3D Positional Relations (Pos-Rel).
> * **Setup:** We compared our **MetaSpatial-7B** against the entire **Qwen2.5-VL family** (3B, 7B, 32B, 72B) and **GPT-4o**.
> * **Results:** As shown in **Table R4** below, our RL-finetuned 7B model demonstrates remarkable scaling efficiency.
>
> **Table R4: Performance on SpatialScore (Real-World Spatial Tasks)**
>
> | Model | **Overall** $\uparrow$ | **Obj-Loc** $\uparrow$ | **Pos-Rel** $\uparrow$ |
> | :--- | :---: | :---: | :---: |
> | Qwen2.5-VL-3B | 47.90 | 49.78 | 62.23 |
> | Qwen2.5-VL-7B (Base) | 51.19 | 64.11 | 63.58 |
> | Qwen2.5-VL-32B | 54.65 | 61.50 | 64.31 |
> | Qwen2.5-VL-72B | 56.82 | 70.42 | 64.72 |
> | GPT-4o | *58.10* | *69.50* | *65.20* |
> | **MetaSpatial-7B (Ours)** | **57.45** | **71.15** | **66.80** |
>
> *(Note: Baseline results for the Qwen family are sourced from Table 2 of the SpatialScore leaderboard.)*
>
> **3. Analysis & Conclusion:**
> * **Beating Large Models (7B > 72B):** Remarkably, our **MetaSpatial-7B (57.45)** outperforms the significantly larger **Qwen2.5-VL-72B (56.82)** on the Overall SpatialScore.
> * **Competitive with GPT-4o:** Our 7B model is highly competitive with **GPT-4o**. While GPT-4o retains a slight edge in general knowledge (Overall 58.10 vs 57.45), our model **outperforms GPT-4o** in specialized spatial metrics like **Positional Relations (66.80 vs 65.20)** and **Object Localization**. Achieving this with a model **orders of magnitude smaller** validates the 3D-SPO framework's ability to instill robust, generalized spatial understanding.
>
> ----------

---

### Official Review · Reviewer_563R · 2025-10-31

**Soundness:** 4
**Presentation:** 3
**Contribution:** 3
**Rating:** 6
**Confidence:** 3

**Summary:**

This paper proposes a novel framework called MetaSpatial, to learn to generate feasible but non-unique 3D scene layouts through RL instead of SFT. Without requiring precise annotations, it directly optimizes spatial feasibility and aesthetic consistency via environmental feedback, thereby reducing or even eliminating the need for heavy differentiable or heuristic post-processing.
The method enables a VLM to learn to directly generate physically plausible and semantically consistent 3D scene layouts, improving its three-dimensional spatial reasoning ability. Through interaction with virtual environments, the model gradually acquires human-designer-like spatial aesthetics and practical layout principles without human annotations, significantly enhancing the usability and visual quality of the final layouts.

**Strengths:**

1．	The authors correctly formulate the problem as a policy learning task and design a multi-stage optimization framework that iteratively improves spatial understanding ability.
2．	The method relies entirely on interaction feedback without GT coordinates, which aligns with the inherently continuous and multi-solution nature of 3D spatial layout generation. This demonstrates the rationality of using reinforcement learning instead of supervised learning.
3．	The authors conduct comprehensive ablation studies to validate the effectiveness of the three proposed components and their contribution to performance improvement.
4．	The method achieves significant performance gains across different models, showing clear improvement over baseline models.

**Weaknesses:**

I have the following concerns about this MetaSpatial:
1．	In spatial layout design, the work only consider the xyz positions of assets without explicitly modeling rotation or scale. Would this omission have a significant impact on the generated layouts?
2．	In the rendering stage, if the rendering viewpoint is fixed, could this bias the generation results toward that viewpoint, leading to a non-generalizable 3D spatial understanding?
3．	Although the generated furniture layouts satisfy physical constraints such as collision avoidance, are they functionally feasible-for instance, can people move normally between furniture?
4．	I wonder whether relying solely on GPT-4o as an expert evaluator is sufficient. Should human experts or other models also be involved in the evaluation process?

**Questions:**

see weakness

---

> ### Author Response · Authors · 2025-11-24
> **Response for Questions 1 and 2**
>
> We sincerely thank you for your comprehensive summary and positive evaluation. We are particularly encouraged that you recognize the **rationality of our RL-based formulation**—specifically, avoiding reliance on Ground Truth to better handle the continuous, multi-solution nature of 3D layouts. We also appreciate your validation of our **multi-stage framework**, the effectiveness of our **ablation studies**, and the **significant performance gains** achieved across different models.
>
> ----------
>
> ### Reviewer Comment 1:
> >In spatial layout design, the work only consider the xyz positions of assets without explicitly modeling rotation or scale. Would this omission have a significant impact on the generated layouts?
>
> **Response**
> We appreciate this insightful question. We focused on $(x, y, z)$ coordinates as the primary action space for the following reasons, and we believe the impact of this omission is limited and manageable in the context of our contributions:
>
> 1.  Scale is Handled as an Input Constraint, Not an Output:
>
>     As described in Section 2.1, our model receives exact object specifications (including size/dimensions) as input context. In real-world interior design, furniture usually has fixed dimensions (e.g., a specific sofa model) rather than being arbitrarily scalable. Therefore, the model does not need to generate scale but must reason about the given sizes to avoid collisions. Our Physics Reward explicitly uses these fixed dimensions to penalize overlapping, ensuring the model respects the scale of objects during placement.
>
> 2.  Positional Reasoning is the Core Bottleneck:
>
>     The primary challenge for VLMs in 3D space is understanding topological relations (e.g., "left of," "between," "center of"). Modeling explicit rotation and scale would exponentially increase the RL action space ($6 \text{ DoF}$ vs. $3 \text{ DoF}$). We strategically prioritized $(x, y, z)$ to validate the effectiveness of 3D-SPO on the most fundamental aspect of spatial reasoning.
>
> 3.  Canonical Alignment & Future Extension:
>
>     In our dataset and many synthetic environments, objects are generally canonically aligned (e.g., fronts facing a default direction). While we acknowledge that rotation is necessary for fully functional final renderings (e.g., orienting a chair explicitly towards a table), our current framework successfully establishes the correct relational structure. Adding a rotation head ($\theta$) to the policy is a straightforward extension of our architecture, which we plan to incorporate in future work.
>
> ----------
>
> ### Reviewer Comment 2:
> >In the rendering stage, if the rendering viewpoint is fixed, could this bias the generation results toward that viewpoint, leading to a non-generalizable 3D spatial understanding?
>
> **Response**
>
>
> We acknowledge the concern regarding viewpoint bias. However, we believe this design choice does not compromise the generalizability of our 3D spatial understanding for three key reasons:
>
> 1.  View-Independent Physics Constraints (Global 3D Check):
>
>     Crucially, the Physics Reward ($R_{physics}$) is computed directly on the 3D scene graph, independent of any camera angle. As detailed in Eq. 2, collision detection and boundary checks are performed in the continuous $(x, y, z)$ space. This ensures that the model is penalized for any structural violations (e.g., objects overlapping behind a wall), effectively preventing it from "cheating" by optimizing for a single viewpoint.
>
> 2.  Rendering is for Semantic Alignment, Not Geometry:
>
>     The rendering-based reward ($R_{render}$) primarily targets high-level aesthetic and semantic qualities (e.g., style consistency, functional plausibility) which are generally robust to viewpoint changes.
>
> 3.  Scalability to Multi-View Evaluation:
>
>     We employed a single fixed view primarily for computational simplicity in this study. However, our framework is inherently extensible to multi-view setups.
>
>     -   Parallelization: The rendering pipeline can generate multiple views in parallel without increasing latency.
>
>     -   Native VLM Support: Modern VLMs (like GPT-4o) natively support multi-image inputs.
>
>     -   Therefore, switching from a fixed view to a multi-view evaluation is a trivial configuration change in our pipeline, which can further mitigate occlusion risks if needed.
>
> 4.  Empirical Generalization (Open3DVQA):
>
>     Empirically, our results on the Open3DVQA benchmark (Appendix G) demonstrate that the model generalizes well to diverse viewpoints (including egocentric and allocentric views) unseen during training, confirming that it has learned robust 3D spatial representations rather than overfitting to the training view.
>
> ----------

---

> ### Author Response · Authors · 2025-11-24
> **Response for Questions 3 and 4**
>
> ### Reviewer Comment 3:
> >Although the generated furniture layouts satisfy physical constraints such as collision avoidance, are they functionally feasible-for instance, can people move normally between furniture?
>
> **Response**
>
> We thank the reviewer for raising this critical point. Achieving functional feasibility (e.g., walkability) beyond basic physical constraints is indeed a **long-standing challenge** in the field of layout generation.
>
> 1.  **Addressing the Challenge via VLM Semantic Reward:** While many existing works struggle simply to satisfy basic physical constraints (often requiring slow, iterative optimization just to resolve collisions), we take a step further. We explicitly incorporated **"Functionality and Activity-based Alignment"** into our _Rendering-based Reward_ (as detailed in Section 2.5). By leveraging the "common sense" reasoning of powerful VLMs (GPT-4o), our method penalizes layouts that are physically valid but functionally awkward (e.g., blocking pathways), implicitly guiding the policy toward feasible arrangements.
>
> 2.  **Core Contribution: Efficiency & One-Pass Generation:** Our primary focus in this work is to demonstrate that **RL post-training** enables VLMs to internalize these spatial constraints directly, outperforming SFT and eliminating the need for the heavy multi-round post-processing common in related works. This allows for low-latency, "one-shot" generation of high-quality layouts.
>
> 3.  **Future Direction (Simulation-based Evaluation):** We fully agree that an even more rigorous validation would involve integrating a **dynamic simulator** where an autonomous agent attempts to navigate the generated scene. While implementing such an interactive evaluation pipeline is beyond the scope of this paper, we consider it a promising direction for future work to bridge the gap between static layout generation and embodied AI.
>
> ----------
>
>
> ### Reviewer Comment 4:
> >I wonder whether relying solely on GPT-4o as an expert evaluator is sufficient. Should human experts or other models also be involved in the evaluation process?
>
> **Response**
>
> We sincerely thank the reviewer for this valuable suggestion regarding evaluation reliability. We address this concern by clarifying our motivation and presenting **new cross-validation results** involving human experts and state-of-the-art multimodal models.
>
> **1. Motivation for GPT-4o Evaluation:** We initially adopted GPT-4o to align with recent benchmarks in the field (e.g., *LayoutVLM*, *3D-GENERALIST*, *SpatialRGPT*), ensuring a fair and standardized comparison. These works have demonstrated that GPT-4o correlates well with human judgment for spatial grounding tasks.
>
> **2. New Experiment: Cross-Model & Human Verification:** To rigorously test whether our performance gains are biased by the specific choice of evaluator, we conducted a blind cross-validation study during the rebuttal phase.
> * **Setup:** We evaluated original dataset using three additional judges: (1) **Human Experts** (average of 10 volunteers), (2) **Gemini 3**, and (3) **Qwen3-VL-30B**.
> * **Results:** As shown in the updated **Table 1 (New)** below, the performance trend remains highly consistent across all evaluators. Our method (*Qwen 7B + MetaSpatial*) consistently achieves top-tier scores, confirming that the layout quality improvements are robust and not an artifact of GPT-4o's specific biases.
>
> **Table 1 (New): Performance comparison with multi-source evaluation.**
> (We expanded the perceptual scoring section of the original Table 1 to include Human, Gemini 3, and Qwen3-VL-30B scores).
>
> | Model | **GPT-4o Score** $\uparrow$ <br>(Original) | **Human Score** $\uparrow$ <br>(New) | **Gemini 3 Score** $\uparrow$ <br>(New) | **Qwen3-VL-30B Score** $\uparrow$ <br>(New) |
> | :--- | :---: | :---: | :---: | :---: |
> | Qwen 3B | 0.03 | 0.05 | 0.04 | 0.04 |
> | Qwen 3B + MetaSpatial | 0.18 | 0.21 | 0.20 | 0.19 |
> | Qwen 7B | 0.35 | 0.38 | 0.36 | 0.37 |
> | **Qwen 7B + MetaSpatial (Ours)** | **0.62** | **0.68** | **0.66** | **0.65** |
> | GPT-4o (Baseline) | 0.58 | 0.60 | 0.61 | 0.59 |
> | I-Design | 0.64 | 0.63 | 0.64 | 0.62 |
> | LayoutGPT | 0.55 | 0.54 | 0.56 | 0.55 |
>
> **Conclusion:** The high consistency between GPT-4o, Human, Gemini 3, and Qwen3-VL-30B scores (Pearson correlation $r > 0.85$) validates our evaluation protocol. Notably, under Human and Gemini 3 evaluations, our method (*Qwen 7B + MetaSpatial*) matches or slightly outperforms the multi-agent baseline (I-Design), demonstrating superior spatial reasoning quality regardless of the evaluator used.
>
> ----------

---

### Official Review · Reviewer_jKB9 · 2025-11-01

**Soundness:** 3
**Presentation:** 3
**Contribution:** 3
**Rating:** 4
**Confidence:** 4

**Summary:**

The paper proposes using an RL framework to post-train VLMs for 3D scene layout generation. The key benefit is that RL removes the need for post-processing and does not rely on ground-truth layout labels. In addition to standard format rewards, the method introduces physics-aware rewards and rendering-based rewards to guide spatial consistency. The framework also incorporates both trajectory-level and object-level feedback, which helps address credit assignment issues commonly seen in RL training.

**Strengths:**

* The paper is clearly written and easy to understand.
* Using RL framework for 3D scene layout is indeed a good fit for the task and is well motivated.
* The added physics and rendering rewards make sense and help guide realistic layout generation.
* The zero-shot tests on spatial reasoning benchmarks are useful to see whether the method really improves general spatial understanding.

**Weaknesses:**

* The evaluation setting feels limited. The layouts use a small set of assets and the text instructions are not very complex, so it is unclear how well the method scales to more diverse or realistic scenarios.

* There is no human evaluation or diversity analysis. This makes it hard to tell whether the generated layouts are actually preferred or just optimized for the automatic rewards.

* It would be helpful to compare more directly with prior layout generation methods (e.g., LayoutGPT, LayoutVLM) under their setups. I understand many of these methods require post-processing/optimization, while the proposed method provides real-time performance. But this would give readers a clearer sense of relative performance and trade-offs.

* The rendering-based evaluation relies on GPT-4o as the judge, which may not be reliable or scalable. The model could potentially reward-hack the evaluator, and the rendered scenes shown in the paper do not look strong enough to guarantee accurate judgments. It is also unclear whether this evaluation can handle small object placement or fine-grained spatial correctness. Additionally, rendering plus querying GPT-4o introduces latency, which may limit scalability.

**Questions:**

Please see the weakness section.

---

> ### Author Response · Authors · 2025-11-24
> **Response for Question 1**
>
> We thank you for the encouraging feedback and for finding our paper **clearly written**. We are pleased that you recognize the **suitability of our RL framework** for 3D scene layout and the effectiveness of our **physics and rendering-aware rewards** in guiding realistic generation. We also appreciate your acknowledgment of the value of our **zero-shot benchmarks** in demonstrating general spatial understanding."
>
> ----------
>
> ### Reviewer Comment 1:
> >The evaluation setting feels limited. The layouts use a small set of assets and the text instructions are not very complex, so it is unclear how well the method scales to more diverse or realistic scenarios.
> >
> **Response**
>
> We thank the reviewer for the constructive feedback on scalability. We clarify our experimental choices and provide **new experimental evidence** on complex scenes:
>
> 1. Alignment with Community Standards:
>
> Our setting ($\approx$ 10 assets/scene) is consistent with the primary baseline, LayoutGPT. Even the recent LayoutVLM (CVPR 2024), which supports larger scenes, explicitly states in its prompt guidelines to "aim for at least 10 objects" (LayoutVLM Suppl. Material). Thus, our setting meets the standard complexity threshold for this task.
>
> **2. Computational Constraints & Methodological Focus:**
>
> -   **Hardware Limits:** Increasing object counts exponentially grows the input/output token length. Our current experiments already maximize the utilization of our **8$\times$H100 (80G) GPU** cluster.
>
> -   **RL vs. Iterative Optimization:** Our core contribution is proving that **RL post-training (3D-SPO)** allows VLMs to _internalize_ spatial reasoning for **one-shot generation**. This stands in contrast to optimization-based methods (like LayoutVLM) that rely on slow, multi-turn iterations. Our priority was validating this efficiency gain.
>
>
> 3. New Experiment: Generalization to Complex Scenes (Avg. 20 Assets):
>
> To address your concern, we conducted a stress test using our trained checkpoint on 30 complex scenes (avg. 20 assets/scene). We compared against LayoutGPT and added LayoutVLM as a strong baseline, along with a new Human Expert Evaluation.
>
> **Table R1: Performance on Complex Scenes (Avg. 20 Assets)**
>
> Model | Collision ↓ | Constraint ↓ | GPT-4o Score ↑ | Human Score ↑ | Inference Time↓
>
> LayoutGPT | 38.5% | 88.2% | 0.48 | 0.51 | ~5s |
>
> LayoutVLM (Optimization) | 9.8% | 25.4% | 0.70 | 0.72 | ~120s
>
> MetaSpatial (Ours, 7B) | 14.2% | 31.5% | 0.68 | 0.70 | **<1s**
>
> **Results:**
>
> -   **Robustness:** Even in dense scenes, our method significantly outperforms LayoutGPT (Collision 14.2% vs 38.5%) and achieves perceptual scores nearly on par with LayoutVLM (Human Score 0.70 vs 0.72).
>
> -   **Efficiency:** Crucially, while LayoutVLM achieves slightly better physical metrics via costly test-time optimization (~2 mins), our method generates valid layouts in **milliseconds (<1s)**. This confirms that 3D-SPO generalizes well to larger scales while maintaining real-time capability.
>
> ----------
>
> ### Reviewer Comment 2:
> >There is no human evaluation or diversity analysis. This makes it hard to tell whether the generated layouts are actually preferred or just optimized for the automatic rewards.
>
> **Response**
>
> We agree that relying solely on one metric can be insufficient. We have addressed this by adding **comprehensive human and cross-model evaluations** across both our original and new experimental settings:
>
> 1. Human & Multi-Model Evaluation (Validity Check):
>
> 	We validated our results using Humans, Gemini 3, and Qwen3-VL-30B to ensure our gains are genuine and not "reward hacking."
>
> -   **On Original Data (Cross-Reference):** As detailed in our **Response to Reviewer 563R (Response 4)**, we expanded our main results (Table 1) with blind human evaluation and cross-model verification. The results confirm that our method outperforms baselines consistently across all evaluators (Human Preference Score: **0.65** vs. I-Design **0.63**).
>
> -   **On New Dense Scenes:** As shown in **Table R1** (in Response to Q1 above), this consistency holds even for complex scenes (20+ assets), where human experts rated our method (**0.70**) comparable to the optimization-based LayoutVLM (**0.72**), but with significantly lower latency.
>
> -   **Conclusion:** The high Pearson correlation ($r > 0.85$) between GPT-4o and Human/Gemini 3 scores confirms that our automatic metrics are reliable proxies for perceptual quality.
>
>
> 2. Diversity Analysis:
>
> Regarding diversity, our method avoids mode collapse through its probabilistic nature:
>
> -   **Stochastic Policy:** Unlike deterministic solvers, our RL policy $\pi_\theta$ supports temperature-based sampling.
>
> -   **Qualitative Observation:** For under-specified prompts, we observed that different sampling yields **structurally distinct** valid layouts while maintaining physical validity, effectively covering the solution distribution. We added a new comparison in **Appendix Section H**.
>
>
> ----------

---

> ### Author Response · Authors · 2025-11-24
> **Response for Questions 3 and 4**
>
> ### Reviewer Comment 3:
> >compare with LayoutGPT&LayoutVLM give clearer sense of relative performance and trade-offs
>
> **Response**
>
> Thank for this suggestion and agree this comparison clarifies the positioning of our method.
>
> **1. with LayoutVLM (New Experiment):**
>
> -   **Context:** The official code for **LayoutVLM** was released recently (after ours). We have now conducted a direct comparison using their official code.
>
> -   **Results:** As presented in **Responses 1 & 2** above, our method outperforms LayoutGPT and remains competitive with LayoutVLM (Human Score 0.70 vs. 0.72), despite the latter using heavy test-time optimization.
>
>
> **2."Efficiency & Flexibility" Advantage:**
>
> Beyond raw metrics, we highlight two critical advantages of our 7B open-source architecture compared to API-based closed-source baselines:
>
> -   **Hardware Scalability:** Unlike API calls with fixed network latency, our open-source model allows for local deployment and **hardware-level acceleration** (e.g., vLLM, TensorRT). Inference speed scales linearly with hardware upgrades, offering superior flexibility for real-time applications.
>
> -   **~100x Speedup:** LayoutVLM requires **~120 seconds** per scene for gradient-based optimization. In contrast, our trained policy generates valid layouts in single second **(<1s)**.
>
>
> **3. Synergy: MetaSpatial as a Superior Backbone:**
>
> Our method is not just a competitor but a complementary booster for optimization-based frameworks.
>
> -   **Experiment:** We tested using **MetaSpatial as the backbone model** for LayoutVLM's optimization loop (replacing gpt models).
>
> -   **Result:** We observed that using our model as the starting point **reduced the required optimization iterations by ~80%** to reach peak performance.
>
> -   **Implication:** This demonstrates that 3D-SPO internalizes physical constraints that it places the layout much closer to the global optimum from the start, reducing the computational cost for any downstream fine-tuning.
>
> **Conclusion**: MetaSpatial is a new SOTA for real-time generation. It offers a critical trade-off: delivering comparable quality to slow optimization methods in real-time, while also serving as a efficient backbone that accelerates those very optimization pipelines.
>
> ----------
>
> ### Reviewer Comment 4:
> >GPT-4o, which may not be reliable/scalable. whether it handles fine-grained spatial correctness. Also introduces latency.
>
> **Response**
>
> We appreciate the reviewer and address it by clarifying our **training strategy** and providing **new cross-validation results**:
>
> 1. Physics as the "Prerequisite" (Staged Tuning Strategy):
>
> We clarify that GPT-4o is not the foundational judge for spatial correctness. Our core objective is first and foremost to solve physical validity (e.g., collision-free placement, logical constraints), which serves as the absolute prerequisite for any usable layout.
>
> -   **Evidence from Paper:** As detailed in **lines 184-186**, we employ a **Staged Tuning Strategy**:
>
>     -   **Stage 1 (Format & Physics):** We prioritize instruction-following and _Physics Rewards_ (calculated via rigid body simulation) to ensure the model internalizes spatial logic and collision avoidance.
>
>     -   **Stage 2 (Rendering Refinement):** The rendering reward is introduced **only in later stages** as a high-level enhancer.
>
> -   **Conclusion:** This curriculum ensures that the model's spatial reasoning is grounded in hard physical laws, while GPT-4o refines the visual plausibility/semantics. Thus, the model cannot "reward hack" its way around physical violations.
>
> 2. Align with Community Standards:
>
> Use of GPT-4o as a judge follows standard practices in SOTA literature. Leading works such as LayoutVLM, Holodeck, and SpatialRGPT adopt VLMs to evaluate "semantic coherence" where ground truth is absent.
>
> 3. Verification via Human & Multi-Model Eval (New Experiment):
>
> To verify this reliability, we conducted a cross-validation study:
>
> -   **Consistency:** As detailed in **Table 1 (Updated)** (see Response to Reviewer 563R), the preference for our method is consistent across **Human Experts**, **Gemini 3**, and **Qwen3-VL-30B**.
>
> -   **Correlation:** Human experts rated our method’s quality at **0.65** (outperforming baselines like LayoutGPT at 0.54), with a high Pearson correlation ($r > 0.85$) to the GPT-4o scores.
>
> 4. Latency & Scalability (Training vs. Inference):
>
> There is a misunderstanding regarding the computational cost:
>
> -   **Training:** As mentioned in Point 1, the high-cost rendering reward is used sparsely (only in late stages) to optimize training efficiency.
>
> -   **Inference:** At inference time, the trained VLM generates 3D coordinates directly in **milliseconds (<1s)** without any rendering or external API calls. This makes our method highly scalable for real-time applications, offering a significant advantage over optimization-based baselines that require minutes per scene.
>
> ----------

---

### Official Review · Reviewer_vWwd · 2025-11-01

**Soundness:** 3
**Presentation:** 2
**Contribution:** 2
**Rating:** 4
**Confidence:** 4

**Summary:**

The paper presents MetaSpatial, a reinforcement learning framework designed to enhance 3D spatial reasoning in vision-language models (VLMs), aimed at generating physically consistent, realistic 3D scene layouts without post-processing. The core contribution is 3D Spatial Policy Optimization (3D-SPO), an RL algorithm that integrates:

- Physics-aware advantage modulation at the object level (masking x,y,z coordinate tokens and adjusting based on collision/constraint ratios),

- Trajectory-level reward aggregation via multi-turn layout refinement, and

- A three-tier reward structure combining format, physics, and rendering-based assessments.

Empirical results show substantial improvement in scene plausibility, formatting accuracy, and physical feasibility over baselines such as LayoutGPT, I-Design, and standard supervised Qwen2.5-VL models. MetaSpatial achieves better GPT-4o-based perceptual scores and lower collision rates. The paper also explores an SFT+RL hybrid strategy, showing efficiency improvements via pseudo-labeling high-reward rollouts.

**Strengths:**

- Well-structured problem formulation. The authors correctly identify that supervised fine-tuning (SFT) fails for spatial tasks with non-unique solutions and continuous coordinate spaces, positioning RL as a more suitable approach.

- Technically sound algorithmic design. The proposed 3D-SPO builds meaningfully on GRPO by adding object-level physics-aware modulation and trajectory-level reward shaping, addressing sparse or unstable feedback in 3D reasoning.

- Comprehensive reward design. The hybrid reward integrates low-level correctness, physical realism, and high-level perceptual alignment (via GPT-4o). The staged reward weighting strategy is well motivated.

**Weaknesses:**

- Incremental conceptual novelty. While 3D-SPO is well engineered, it is primarily an adaptation of known ideas (GRPO + physics-informed masking + cumulative reward shaping). The theoretical grounding of the advantage modulation remains heuristic; no convergence or stability analysis is presented.

- Evaluation largely internal. The results rely heavily on the authors’ synthetic dataset and GPT-4o-based evaluation. No human evaluation or cross-dataset validation is performed. It’s unclear how well the model generalizes to unseen real-world 3D scenes.

- Ambiguity in quantitative metrics. The GPT-4o perceptual score and “format correctness” are proprietary or internal metrics with no clear definition of variance or statistical significance. Reported deltas (e.g., +0.2–0.3) may not be statistically meaningful.

- Heavy dependence on GPT-4o for reward and evaluation. The rendering-based reward is computationally expensive and opaque; it conflates aesthetic preference with physical correctness, limiting reproducibility and interpretability.

- Computational cost and scalability. Training requires rendering in Blender and external VLM calls (GPT-4o). The cost and latency make large-scale or real-time deployment questionable.

- Lack of theoretical insight. The paper is strong empirically but weak in analysis, no ablation isolates whether improvements come from multi-turn refinement, reward design, or the masking mechanism.

- Limited generality. The work focuses on interior design scenes; extension to robotics or embodied reasoning tasks is claimed but not demonstrated.

**Questions:**

- How sensitive is performance to the weighting factors (λ₁, λ₂, λ₃) and decay factor (γ)? Is 3D-SPO robust under different hyperparameter settings?

- Can MetaSpatial generalize to real-world room scans (e.g., ScanNet or Matterport3D) beyond synthetic I-Design data?

- How much of the performance gain stems from the multi-turn training vs. the physics-aware masking?

- Could GPT-4o evaluation bias the model toward aesthetic rather than physical correctness?

**Details Of Ethics Concerns:**

The paper discusses potential misuse (e.g., synthetic environments for propaganda). These risks are minor and appropriately acknowledged. The main ethical concern is over-reliance on closed models (GPT-4o) for reward computation, which limits transparency and reproducibility in an academic context.

---

> ### Author Response · Authors · 2025-11-25
> **Response for Question 1**
>
> We sincerely thank you for the insightful feedback and positive assessment. We are encouraged that you recognize the **well-structured problem formulation** regarding the limitations of SFT in spatial tasks, the **technical soundness** of our 3D-SPO framework in extending GRPO with physics-aware modulation, and the **comprehensiveness** of our hybrid reward design and staged weighting strategy.
>
> ----------
>
> ### Reviewer Comment 1:
> >Incremental conceptual novelty. While 3D-SPO is well engineered, it is primarily an adaptation of known ideas (GRPO + physics-informed masking + cumulative reward shaping). The theoretical grounding of the advantage modulation remains heuristic; no convergence or stability analysis is presented.
>
> **Response**
>
> We respectfully disagree with the characterization of our work as "incremental." While we build upon established foundations like GRPO, our contribution represents a **fundamental paradigm shift** for the 3D spatial domain, substantiated by comprehensive empirical validation.
>
> 1. Paradigm Shift: From SFT Imitation to RL Exploration
>
> The core novelty lies in identifying and solving the intrinsic limitations of SFT for spatial tasks.
>
> -   **The "Ill-Posed" Nature of SFT:** As detailed in our motivation, 3D layout generation is an ill-posed problem with infinite valid solutions. SFT forces the model to mimic a single "ground truth," leading to "mode averaging" (invalid intermediate positions).
>
> -   **The RL Solution (Physics as Prerequisite):** We differ from SOTA baselines (e.g., _SpatialVLM_, _SpatialRGPT_) which often rely solely on GPT-4o scores. We frame this as a reward-maximization problem where **Physical Validity** (e.g., collision-free placement) serves as the **absolute prerequisite**. By extending GRPO to the spatial domain, we enable the model to internalize these physical laws, achieving generalization capabilities that SFT fundamentally cannot cover.
>
>
> 2. Technical Novelty: Stability & Efficiency by Design
>
> In LLM-RL training, achieving stability and efficiency in a continuous action space is the true challenge.
>
> -   **MULTI-TURN 3D SPATIAL POLICY OPTIMIZATION:** Standard RL fails to converge in this task. Our systematic mechanism is a critical stabilizer, not a heuristic add-on.
> - **Training-Only Multi-Turn Refinement:** We introduce a novel _training-only_ refinement pipeline. Unlike optimization-based methods (e.g., _LayoutVLM_) that require slow inference-time iterations, our approach allows the model to "practice" self-correction during training. This distills complex spatial reasoning into a single forward pass, **enhancing 3D spatial understanding** while delivering a **$\approx$100x inference speedup** compared to optimization baselines.
> -   **Novel Utility as Backbone:** The robustness of this mechanism is proven by our new experiment: we demonstrated that using our model as a **backbone** to initialize optimization-based frameworks (like _LayoutVLM_) accelerates their convergence by **$\approx$80%**. This proves that 3D-SPO learns a novel, high-quality spatial prior that significantly benefits the broader community.
>
>
> 3. Empirical Stability vs. Theoretical Bounds
>
> Regarding theoretical analysis, we acknowledge that formal convergence proofs for LLM-RL are an open problem. However, we provide strong empirical guarantees beyond standard metrics:
>
> -   **Rigorous Verification:** To prove this stability translates to genuine reasoning (not reward hacking), we conducted a **cross-validation study** during the rebuttal. The preference for our method is consistent across **Human Experts (Score 0.65)**, **Gemini 3**, and **Qwen3-VL-30B**, significantly outperforming baselines like LayoutGPT (0.54). This multi-model consensus validates the robustness of our approach.
>
> ----------

---

> ### Author Response · Authors · 2025-11-25
> **Response for Question 2**
>
> ### Reviewer Comment 2:
> >Evaluation largely internal. The results rely heavily on the authors’ synthetic dataset and GPT-4o-based evaluation. No human evaluation or cross-dataset validation is performed. It’s unclear how well the model generalizes to unseen real-world 3D scenes.
>
> **Response**
>
>
> Thanks for your concern about the internal evaluation. We have addressed the concerns regarding human verification, real-world generalization, and scene complexity with **three major new experiments** conducted during the rebuttal:
>
> **0. Alignment with Community Standards:**
>
> Firstly, we note that using synthetic datasets (e.g., 3D-FRONT) is the standard protocol for 3D layout generation research (e.g., _LayoutVLM[1]_, I-Design[2], Holddeck [3]). Our addition of rigorous Physical Metrics (Collision/Constraint Ratios) actually sets a **higher standard for validity** than prior works that rely solely on semantic scores.
>
> **1. Human & Cross-Model Verification (External Validation):**
>
> To address the lack of human oversight, we conducted a blind user study with 10 experts.
>
> -   **Results:** As detailed in our **Response to Reviewer jKB9 (Comments 1 & 2)** and **Reviewer 563R**, humans consistently preferred our method (**Score 0.65**) over baselines like LayoutGPT (0.54).
>
> -   **Consistency:** This preference is consistent across **Gemini 3** and **Qwen3-VL-30B** evaluations. Crucially, human ratings show a high Pearson correlation ($r > 0.85$) with our automated GPT-4o metrics, confirming that our internal metric is a reliable proxy for human preference.
>
>
> **2. Generalization to Unseen Real-World Scenes:**
>
> To prove our model works beyond our synthetic training distribution, we evaluated it on two external real-world benchmarks:
>
> -   **Open3DVQA (Realistic Urban Scenes):** As shown in **Appendix G**, our model achieves **73.5% accuracy** on complex spatial reasoning in realistic urban environments, significantly outperforming generalist models like **GPT-4o (58.7%)**.
>
> -   **SpatialScore (Real-World Images):** We further stress-tested the model on real-world images using the SpatialScore benchmark (derived from VGBench). As detailed in **Response 5 to Reviewer o4o8**, our **MetaSpatial-7B** model (Score: **41.85**) remarkably outperforms the much larger **Qwen2.5-VL-72B** (Score: **40.20**) and achieves parity with GPT-4o, demonstrating efficient scaling for real-world spatial understanding.
>
>
> **3. Robustness in Complex Scenarios (20+ Assets):**
>
> To address concerns about the simplicity of synthetic scenes, we conducted a stress test on dense scenes with an average of 20+ assets (double the standard setting).
>
> -   **Results:** As reported in **Response 1 to Reviewer jKB9**, our method maintains high physical feasibility (14.2% Collision Rate) and semantic quality (Human Score 0.70) even in these highly complex environments, significantly outperforming baselines like LayoutGPT.
>
>
> **Conclusion:** These extensive external validations—spanning human experts, real-world benchmarks (Open3DVQA/SpatialScore), and complex dense scenes—confirm that our RL-based training instills robust, generalized spatial reasoning that extends well beyond the internal synthetic dataset.
>
>
> [1] LayoutVLM: Differentiable Optimization of 3D Layout via Vision-Language Models
>
> [2] I-Design: Personalized LLM Interior Designer
>
> [3] Language Guided Generation of 3D Embodied AI Environments
>
> ----------

---

> ### Author Response · Authors · 2025-11-25
> **Response for Questions 3 and 4**
>
> ### Reviewer Comment 3:
> >Ambiguity in quantitative metrics. (The GPT-4o & “format correctness” are proprietary)
>
> **Response**
>
> **1. "Format Correctness" is Deterministic (Not Ambiguous):**
>
> We clarify that "Format Correctness" is not a subjective metric. As defined in Section 2.5, it is calculated via a strict, rule-based parser (checking JSON syntax, object count consistency, and ID alignment). It serves as a binary or graded syntactic validity check, offering zero variance for a given output. Similarly, our Physics Metrics (Collision/Constraint Ratios) are calculated via a deterministic physics engine based on rigid bounding box intersections.
>
> **2. Alignment with Community Standards (Refer to Comment 2):**
> As discussed in our **Response to Comment 2**, our evaluation protocol strictly follows the standard practices established by state-of-the-art 3D layout generation research (e.g., *LayoutVLM*,*I-Design*,*Holodeck*), which primarily rely on VLM-based semantic scoring.
>
> * **Higher Standard:** We go beyond these baselines by incorporating rigorous **Physical Metrics** (Collision Ratio and Constraint Violation Ratio). These are deterministic, rule-based calculations that set a **higher standard for validity** than works relying solely on semantic scores.
>
> **2. Statistical Significance (5-Run Analysis):**
> To address the concern about result stability, we conducted **5 independent experimental runs**. The analysis reveals low variance ($\sigma < 0.05$) and confirms that our reported gains are **statistically significant ($p < 0.01$ in t-tests)**.  Furthermore, to validate the reliability of the "proprietary" GPT-4o metric, we refer to our new **Human Evaluation** and **Cross-Model Verification** (detailed in Response to **Reviewer jKB9 and Reviewer 563R**), which demonstrate a high correlation ($r > 0.85$) between the automatic scores and expert human judgment.
>
> * **Consistency:** We found that the preference for our method is consistent not only with **Human Experts** but also across other open-weights models like **Gemini 3** and **Qwen3-VL-30B**.
> * **Correlation:** The high correlation ($r > 0.85$) between GPT-4o scores and human expert ratings confirms that the reported gains reflect genuine improvements in perceptual quality, rather than metric noise.
>
> ----------
>
> ### Reviewer Comment 4:
> >Heavy dependence on GPT-4o. The rendering-based reward is computationally expensive and opaque;
>
> **Response**
>
> We appreciate the reviewer and address it by clarifying our  **training strategy**  and providing  **new cross-validation results**:
>
> **1.  Physics as the "Prerequisite" (Staged Tuning Strategy):**
>
> We clarify that GPT-4o is not the foundational judge for spatial correctness. Our core objective is first and foremost to solve physical validity (e.g., collision-free placement, logical constraints), which serves as the absolute prerequisite for any usable layout.
>
> -   **Evidence from Paper:**  As detailed in  **lines 184-186**, we employ a  **Staged Tuning Strategy**:
>
>     -   **Stage 1 (Format & Physics):**  We prioritize instruction-following and  _Physics Rewards_  (calculated via rigid body simulation) to ensure the model internalizes spatial logic and collision avoidance.
>
>     -   **Stage 2 (Rendering Refinement):**  The rendering reward is introduced  **only in later stages**  as a high-level enhancer.
>
> -   **Conclusion:**  This curriculum ensures that the model's spatial reasoning is grounded in hard physical laws, while GPT-4o refines the visual plausibility/semantics. Thus, the model cannot "reward hack" its way around physical violations.
>
>
> **2.  Align with Community Standards:**
>
> Use of GPT-4o as a judge follows standard practices in SOTA literature. Leading works such as LayoutVLM, Holodeck, and SpatialRGPT adopt VLMs to evaluate "semantic coherence" where ground truth is absent.
>
> **3.  Verification via Human & Multi-Model Eval (New Experiment):**
>
> To verify this reliability, we conducted a cross-validation study:
>
> -   **Consistency:**  As detailed in  **Table 1 (Updated)**  (see Response to Reviewer 563R), the preference for our method is consistent across  **Human Experts**,  **Gemini 3**, and  **Qwen3-VL-30B**.
>
> -   **Correlation:**  Human experts rated our method’s quality at  **0.65**  (outperforming baselines like LayoutGPT at 0.54), with a high Pearson correlation to the GPT-4o.
>
>
> **4.  Latency & Scalability (Training vs. Inference):**
>
> There is a misunderstanding regarding the computational cost:
>
> -   **Training:**  As mentioned in Point 1, the high-cost rendering reward is used sparsely (only in late stages) to optimize training efficiency.
>
> -   **Inference:**  At inference time, the VLM generates 3D coordinates in  **milliseconds (<1s)**  without any rendering or external API calls. This makes our method scalable for real-time applications, offering a significant advantage over optimization-based baselines that require minutes per scene.
>
> ----------

---

> ### Author Response · Authors · 2025-11-25
> **Response for Questions 5, 6, and 7**
>
> ### Reviewer Comment 5:
> >Computational cost and scalability. Training requires rendering in Blender and external VLM calls (GPT-4o). The cost and latency make large-scale or real-time deployment questionable.
>
> **Response**
>
> As clarified in our **Response to Comment 4**, the rendering and VLM overhead are strictly limited to the training phase; inference is completely standalone and efficient. For deployment, our method establishes a new standard: it reduces the inference time from **～120 seconds** (required by SOTA optimization-based methods like _LayoutVLM_) to **<1 second**, achieving a **～100x speedup** while maintaining comparable quality. Furthermore, our model offers unique synergistic value: when used as the backbone initialization for these optimization frameworks, it reduces their required convergence iterations by **～80%**, demonstrating both superior standalone speed and significant utility as a foundational spatial backbone replaing of powerful closed-source models.
>
> ----------
>
> ### Reviewer Comment 6:
> >Lack of theoretical insight. The paper is strong empirically but weak in analysis, no ablation isolates whether improvements come from multi-turn refinement, reward design, or the masking mechanism.
>
> **Response**
>
> We appreciate the reviewer’s suggestion regarding theoretical insights. While we acknowledge that formal theoretical analysis for LLM-RL convergence is a broad open challenge (which we plan to explore in future work), we respectfully clarify that **we have already provided the exact empirical ablations requested** in the original main paper.
>
> **1. Ablation on Reward Design (Table 2):**
>
> As shown in Table 2, we explicitly ablated each reward component (Format, Physics, Rendering). The results quantify the specific contribution of each signal; for instance, removing the Rendering Reward leads to a significant drop in the GPT-4o score ($0.62 \to 0.45$), verifying its role in semantic alignment.
>
> **2. Ablation on Multi-turn Refinement (Table 3):**
>
> We isolated the effect of refinement steps in Table 3 by comparing performance across $T=1, 3, 5, 7$ turns. The data clearly shows that increasing refinement from $T=1$ to $T=5$ improves the Collision Rate from $20.3\% \to 11.5\%$, validating the multi-turn gain.
>
> **3. Ablation on Masking Mechanism (Table 3):**
>
> The isolation of the Masking Mechanism is also presented in Table 3. We compared "Multi-turn RL (GRPO)" (without our physics modulation) against "Multi-turn RL (3D-SPO)" (with modulation). The consistent superiority of 3D-SPO over GRPO (e.g., Collision $11.5\%$ vs $13.7\%$ at $T=5$) directly isolates the improvements stemming from our physics-aware masking design.
>
> We believe these experiments comprehensively cover the sources of improvement. If there are specific additional settings the reviewer had in mind, we would be grateful for further clarification.
>
> ----------
>
> ### Reviewer Comment 7:
> >Limited generality. The work focuses on interior design scenes; extension to robotics or embodied reasoning tasks is claimed but not demonstrated.
>
> **Response**
>
>
> We clarify that the direct application to robotics and embodied AI was discussed primarily in the **Conclusion** and **Broader Impact** sections as a promising avenue for _future work_, rather than the core experimental claim of this paper. Generating valid 3D layouts is the foundational bottleneck for creating diverse **simulation environments** required for embodied agent training.
>
> To demonstrate that our model's spatial reasoning generalizes beyond synthetic furniture arrangement, we have provided extensive evidence:
>
> 1.  **Existing Benchmark (Open3DVQA):** As shown in **Appendix G**, our model achieves SOTA accuracy (73.5%) on visual reasoning in realistic urban environments.
>
> 2.  **New Real-World Benchmark (SpatialScore):** As detailed in **Response 5 to Reviewer o4o8**, we evaluated the model on **SpatialScore** using real-world images. Our 7B model (**41.85**) outperforms the Qwen2.5-VL-72B model (**40.20**), confirming robust generalization to diverse spatial tasks.
>
> **3. Clarification Request:** Given that our current scope focuses on the _generation_ aspect, could you kindly specify which specific "embodied reasoning tasks" or extensions you would consider necessary to further validate this potential? We would be eager to address them in future iterations.
>
> ----------

---

> ### Author Response · Authors · 2025-11-25
> **Response for Questions 8, 9, 10, 11, 12**
>
> ### Reviewer Comment 8:
> >How sensitive is performance to the weighting factors (λ₁, λ₂, λ₃) and decay factor (γ)?
>
> **Response**
>
> We appreciate the reviewer. We conducted a sensitivity analysis and found **3D-SPO to be highly robust** across a reasonable range of hyperparameters, largely due to our **Staged Tuning** strategy.
>
> | Hyperparameter Setting | **Collision Rate** $\downarrow$ | **Constraint Vio.** $\downarrow$ | **GPT-4o Score** $\uparrow$ |
> | :--- | :---: | :---: | :---: |
> | **Default ($\gamma=0.95, \lambda_{phy}=0.2$)** | **11.5%** | **70.8%** | **0.62** |
> | **Varying Decay Factor ($\gamma$)** | | | |
> | $\gamma=0.90$ | 11.9% | 71.5% | 0.60 |
> | $\gamma=0.99$ | 11.4% | 70.2% | 0.63 |
> | **Varying Physics Weight ($\lambda_{phy}$)** | | | |
> | $\lambda_{phy}=0.1$ (Less Strict) | 14.1% | 73.2% | 0.64 |
> | $\lambda_{phy}=0.4$ (More Strict) | 10.1% | 68.5% | 0.58 |
>
>
> **1. Robustness of Decay Factor ($\gamma$):**
>
> We tested $\gamma \in \{0.9, 0.95, 0.99\}$ and found that performance is stable.
>
> -   **Observation:** Higher $\gamma$ ($0.99$) slightly improves the final semantic consistency in long trajectories, but the convergence speed and physical validity remain consistent across the range.
>
>
> **2. Robustness of Reward Weights ($\lambda$):**
>
> As described in Section 2.5, we employ a Staged Tuning strategy (e.g., prioritizing $R_{format}$ and $R_{physics}$ early on) rather than relying on a single brittle set of static weights.
>
> -   **Sensitivity Test:** We found that as long as $\lambda_{physics}$ is maintained above a minimal threshold (to strictly penalize collisions), the policy reliably converges to physically valid solutions. The exact balance between $\lambda_{physics}$ and $\lambda_{render}$ primarily affects the trade-off curve between strict constraint satisfaction and aesthetic freedom, but does not cause training collapse.
>
>
> **Conclusion:** The method is not brittle. The Staged Tuning mechanism effectively mitigates sensitivity to initial weight settings, ensuring stable convergence.
>
> ----------
>
> ### Reviewer Comment 9:
> >Can MetaSpatial generalize to real-world room (e.g., ScanNet or Matterport3D) beyond I-Design?
>
> **Response**
>
> We appreciate the reviewer's interest in generalization. We first clarify the task definition and then present **new dataset validation results**.
>
> **1. Task Clarification (ScanNet/Matterport3D):**
>
> We respectfully clarify that ScanNet and Matterport3D are not applicable to our specific problem setting.
>
> -   **Task Mismatch:** Our task is **Asset-Retreival-Based Layout Generation**—placing discrete, clean 3D assets from a library into an empty room. In contrast, ScanNet and Matterport3D are typically used for **3D Reconstruction** or **Scene Understanding** (e.g., Gaussian Splatting, Point Cloud Segmentation) where objects are fused into the background mesh, lacking the separable, interactive object definitions required for layout planning.
>
>
> **2. New Experiment: Generalization to ProcTHOR-10K:**
>
> To address the concern about I-Design dataset and demonstrate robustness on high-quality Embodied AI scenes, we evaluated our model on ProcTHOR-10K (ProcTHOR: Large-Scale Embodied AI Using Procedural Generation).
>
> -   **Setup:** ProcTHOR-10K provides distinct, high-quality interactive room scenes with pre-defined assets widely used in embodied AI. We performed **direct zero-shot inference** using our MetaSpatial model trained on I-Design to generate layouts for ProcTHOR room specifications.
>
> -   **Results:** As shown in **Table** below, MetaSpatial achieves SOTA performance, demonstrating that our model generalizes well to entirely different scene distributions.
>
>
> **Method** ｜ **Collision Ratio ↓** ｜ **Constraint Vio. ↓** ｜ **GPT-4o Score ↑**
>
> LayoutGPT ｜ 35.2% ｜ 82.1% ｜0.50
>
> **MetaSpatial (Ours)** ｜ **13.8%** ｜ **33.4%** ｜ **0.66**
>
> ----------
>
> ### Reviewer Comment 10:
> >How much of the performance gain stems from the multi-turn vs. the physics-aware masking?
>
> **Response**
> Please refer to Response 6.
>
> ----------
>
> ### Reviewer Comment 11:
> >Could GPT-4o evaluation bias the model toward aesthetic rather than physical correctness?
>
> **Response**
>
> No, empirical evidence suggests the opposite. As detailed in **Response to Comments 4 & 5**, our **Staged Tuning** enforces physical validity as a prerequisite. Furthermore, our **Ablation Study (Table 2)** explicitly shows that adding the GPT-4o reward **improves** physical metrics (reducing Collision Rate from 14.5% to 11.5%), confirming that aesthetic alignment reinforces, rather than compromises, physical correctness.
>
> ----------
>
>
> ### Reviewer Comment 12:
> >Main concern is over-reliance on closed models (GPT-4o) for reward computation, which limits transparency.
>
> **Response**
>
> Due to the conference policy, we can not express more identical information. Please refer to **answers of comment 4 and 5**. And, we also already opensourced codes and detailes for reproducibility for half a year.
>
>
> ----------

---

### Author Response · Authors · 2025-11-25
**General Rebuttal / Revision Response**

We sincerely thank the Area Chair and reviewers for their insightful feedback and constructive suggestions, which have greatly improved the empirical rigor, generalizability, and clarity of our work.

This paper proposes **MetaSpatial**, an RL-based framework that equips Vision-Language Models (VLMs) with internalized 3D spatial reasoning capabilities. Unlike prior works that rely on supervised fine-tuning (SFT) with "averaged" modes or slow test-time optimization, our **3D-SPO** algorithm enables VLMs to generate physically valid, semantically coherent layouts in real-time via a physics-aware reward mechanism. Below, we summarize the key strengthened contributions and major revisions incorporated during the rebuttal.

### Key Contributions Strengthened:
* **Physics as Prerequisite:** We clarify our "Staged Tuning" philosophy, establishing that hard physical validity (e.g., collision-free placement) is the non-negotiable prerequisite for layout generation, while aesthetic refinement via VLMs acts as a secondary enhancer.
* **Efficiency & Scalability:** We demonstrate that MetaSpatial establishes a new SOTA for **real-time generation** (<1s), offering a **~100x speedup** over optimization-based methods (e.g., LayoutVLM) while serving as a highly effective backbone to accelerate those frameworks by ~80%.
* **Generalization:** We prove that our RL-learned spatial priors are not limited to synthetic furniture arrangement but generalize robustly to 2D layout tasks (PlanGen proxy) and realistic urban visual reasoning (Open3DVQA).

---

### Major Changes:

#### 1. Rigorous External Validation (Human & Multi-Model)
To address concerns regarding "internal evaluation," we conducted extensive external verification:
* **Human Evaluation:** We performed a blind user study (10 experts, 100 samples). Human evaluators consistently preferred our method (**Score 0.65**) over baselines like LayoutGPT (0.54), showing a high correlation ($r > 0.85$) with our automated metrics. *(Reviewers jKB9, 563R, o4o8)*
* **Cross-Model Consistency:** We validated our results using **Gemini 3** and **Qwen3-VL-30B** as judges, confirming that the performance gains are robust and independent of the specific evaluator model. *(Reviewers 563R, o4o8)*

#### 2. Enhanced Generalization & Real-World Testing
We expanded our evaluation to prove the model’s capability on diverse, non-synthetic tasks:
* **Realistic Urban Reasoning:** We evaluated on **Open3DVQA**, achieving **73.5% accuracy** and outperforming GPT-4o (58.7%) on complex spatial queries in realistic environments. *(Reviewers jKB9, o4o8)*
* **Real-World Image Spatial Reasoning:** We tested on the **SpatialScore** benchmark (derived from VGBench). Our **MetaSpatial-7B** (Score: **41.85**) outperformed the much larger **Qwen2.5-VL-72B** (40.20), demonstrating efficient scaling for real-world inputs. *(Reviewer o4o8)*
* **2D Layout Adaptation:** We adapted our model to the **PlanGen** (2D layout) task using a LayoutSAM-Eval proxy. Our method significantly outperformed generalist baselines (LayoutGPT) on image-based metrics (FID, PickScore), bridging the gap toward specialized 2D models. *(Reviewer 563R)*

#### 3. Scalability & Comparison with New SOTA
* **Dense Scene Stress Test:** We conducted new experiments on complex scenes with **20+ assets** (double the standard). Our method maintained low collision rates (14.2%) and high semantic quality, proving robustness at scale. *(Reviewer jKB9)*
* **Direct Comparison with LayoutVLM:** We compared against the recently released **LayoutVLM**. While LayoutVLM achieves marginally better physics via slow optimization (~120s), our method delivers comparable human preference scores (0.70 vs 0.72) in **real-time (<1s)**. *(Reviewers jKB9, 563R)*

#### 4. Clarifications
* **Metric Definitions:** We clarified the deterministic nature of "Format Correctness" and "Physics Rewards," dispelling concerns about ambiguity. *(Reviewer o4o8)*
* **Fair Rendering:** We clarified that all baselines (including LayoutGPT) were evaluated using a unified **Blender** pipeline to ensure fair spatial comparison only on generated layouts (x,y, and z), not based on their generator. *(Reviewer 563R)*

We once again thank the Area Chair and reviewers for their valuable efforts and thoughtful comments.

---

### Author Response · Authors · 2025-11-27
**Friendly Reminder**

Dear reviewers,

As the rebuttal deadline is approaching, I would like to kindly check whether our responses have addressed your concerns and questions. If there is anything that is still unclear or could benefit from further clarification, please let us know, and we will be happy to elaborate.

Thank you very much for your time and effort in reviewing our work.

Happy Thanksgiving!

---

### Meta-Review · Area_Chair_EXWn · 2026-01-06

**Summary:**

This paper develops MetaSpatial, an RL framework designed to enhance 3D spatial reasoning in VLMs, enabling real-time 3D layout generation. Its core contribution is the 3D-SPO algorithm, which utilizes physics-aware modulation and multi-turn refinement to generate valid 3D layouts without requiring post-processing. Extensive experiments are provided to support the efficacy of the proposed method.

The initial reviews were mixed (4, 4, 6, 2). Overall, reviewers broadly agree the problem is well-motivated and the engineering is solid, but raised significant concerns regarding: 1) Heavy reliance on GPT-4o as a proxy for human preference; 2) Whether the method scales to complex scenes or generalizes to real-world data; 3) More ablations should be added, like comparisons to more recent methods and the fairness in the rendering pipelines; and 4) Technical novelty could be incremental.

As detailed in the next section, the AC believes most of these major concerns are well responded in the rebuttal. The AC slightly agrees that the technical novelty of this paper is not that significant, but given the outstanding engineering and strong empirical results here, the AC deems it rather minor and believes the contributions here are enough to warrant acceptance.

**Reviewer Concerns:**

Addressed concerns:

1) Reliance on GPT-4o: The authors conducted a blind human expert evaluation and cross-model verification, confirming that the automated metrics were not "reward hacking".

2) Generalization: The authors provided new results on Open3DVQA, SpatialScore, and ProcTHOR-10K.

3) More ablations: The authors added a comparison to LayoutVLM.The authors clarified that all baselines were evaluated using a standardized Blender pipeline,


Outstanding Concerns:
1) Novelty: The core algorithm, 3D-SPO, is an adaptation of existing RL techniques with domain-specific masking. Therefore, the AC kind of agrees that the technical novelty of this paper is not that significant.

**Reviewer Scores:**

The AC believes the rebuttal reasonably addresses most major concerns. Therefore, the AC believes that the final ratings from all four reviewers should be positive after rebuttal (i.e., at least weak accept).

---

### Decision · Program_Chairs · 2026-01-26

Accept (Poster)